# ONE LANGUAGE, MANY GAPS: EVALUATING DIALECT FAIRNESS AND ROBUSTNESS OF LARGE LANGUAGE MODELS IN REASONING TASKS

## ABSTRACT

Language is not monolithic. While many benchmarks are used as proxies to systematically estimate Large Language Models' (LLM) performance in real-life tasks, they tend to ignore the nuances of within-language variation and thus fail to model the experience of speakers of minority dialects. Focusing on African American Vernacular English (AAVE), we present the first study on LLMs' fairness and robustness to a dialect in canonical reasoning tasks (algorithm, math, logic, and comprehensive reasoning). We hire AAVE speakers, including experts with computer science backgrounds, to rewrite seven popular benchmarks, such as HumanEval and GSM8K. The result of this effort is **ReDial**, a dialectal benchmark comprising $1.2K+$ parallel query pairs in Standardized English and AAVE. We use ReDial to evaluate state-of-the-art LLMs, including GPT-4o/4/3.5-turbo, LLaMA-3.1/3, Mistral, and Phi-3. We find that, compared to Standardized English, **almost all of these widely used models show significant brittleness and unfairness to queries in AAVE**. Furthermore, AAVE queries can degrade performance more substantially than misspelled texts in Standardized English, even when LLMs are more familiar with the AAVE queries. Finally, asking models to rephrase questions in Standardized English does not close the performance gap but generally introduces higher costs. Overall, our findings indicate that LLMs provide unfair service to dialect users in complex reasoning tasks. Code can be found at https://anonymous.4open.science/r/redial_eval-0A88.

## 1 INTRODUCTION

Over the last few decades, linguistics has firmly established that language varies along different external dimensions such as geography, age, and gender, *dialectal* variation being among the most perspicuous manifestations (Chambers & Trudgill, 1998). Speakers of 'non-standard' dialects are known to experience implicit and explicit forms of discrimination in everyday situations, including housing, education, work, and criminal justice (Baugh, 2005; Adger et al., 2014; Rickford & King, 2016; Drożdżowicz & Peled, 2024). As Large Language Models (LLMs) are increasingly employed as a service and by a rapidly growing user base (Milmo, 2023; La Malfa et al., 2024), it is vital to understand the service quality that they provide to different groups and demographics.

In this work, we examine LLMs' **dialect robustness and fairness**. Previous studies have shown that language models exhibit biases to dialect prompts in tasks such as hate speech detection and reading comprehension (Sap et al., 2019; Ziems et al., 2023), as well as making judgments about employability and criminal justice (Hofmann et al., 2024). Equally relevant, yet less studied, are tasks that require reasoning abilities for problem-solving, decision-making, and critical thinking (Wason, 1972; Huth, 2004; Huang & Chang, 2022; Qiao et al., 2022). For instance, algorithm-related tasks (e.g., generation, debugging, etc.) figure prominently in real user queries, as reflected by their first place on the ArenaHard quality board (Li et al., 2024) and their third place on the WildChat frequency board (Zhao et al., 2024). However, existing dialectal benchmarks (e.g., Ziems et al., 2023) do not cover these tasks, and current popular reasoning benchmarks such as HumanEval (Chen et al., 2021) and GSM8K (Cobbe et al., 2021) are constructed in Standardized English. It is thus unclear whether LLMs are **fair** when responding to reasoning tasks expressed in 'non-standard' English dialects. Moreover, dialect queries can also be used to test LLMs' **robustness**. Adversarial robustness

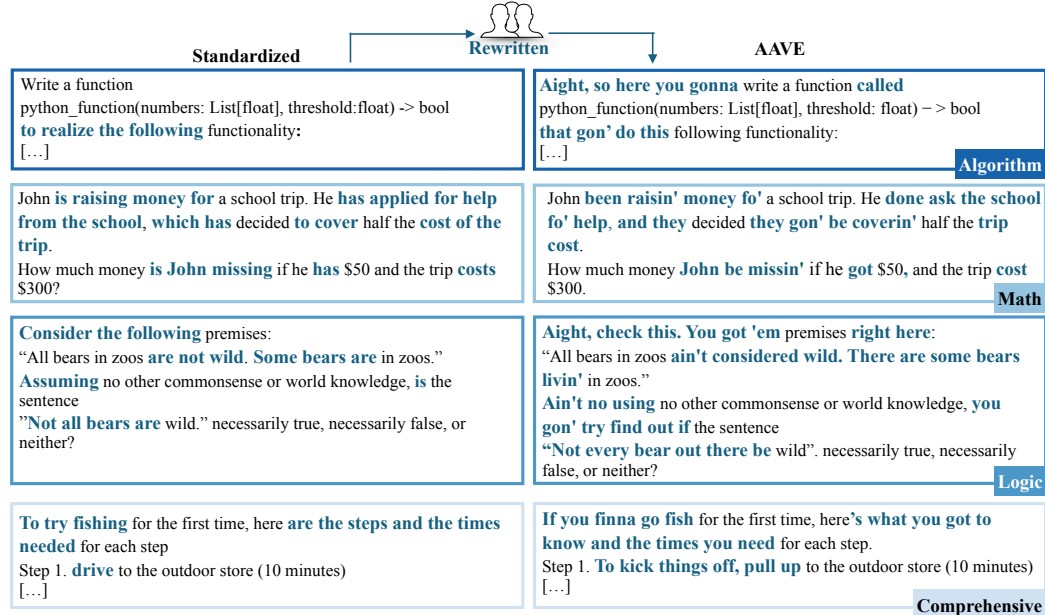

Figure 1: ReDial is a dialect reasoning benchmark composed of $1.2K+$ Standardized English-AAVE parallel queries. Its source data comes from existing benchmarks in Standardized English. AAVE speakers are hired to rewrite each instance in their dialect but preserve their original intent, meaning, and ground truth output label to form their AAVE counterparts.

provides a consolidated framework to test LLMs on slight variations of existing tasks (Moradi & Samwald, 2021; Jin et al., 2023). In this sense, dialects reformulate a problem while maintaining its semantics, i.e., they test what has been referred to as *semantic robustness* (Malfa & Kwiatkowska, 2022).

In this work, we present the first study on evaluating LLMs in reasoning tasks expressed in African American Vernacular English (AAVE), with the objective to evaluate LLMs' fairness and robustness towards a dialect. We choose AAVE since around 33 million people worldwide and approximately 80% of African Americans in the United States speak AAVE, with reports of discriminative behaviors in various scenarios (Lippi-Green, 1997; Purnell et al., 1999; Massey & Lundy, 2001; Grogger, 2011; Rickford & King, 2016). Our study aims to understand whether LLMs hold biases against AAVE speakers in reasoning tasks. Previous approaches in creating AAVE benchmarks from existing Standardized English data either (i) primarily use validated lexical and morphosyntactic transformation rules (Ziems et al., 2022; 2023), which fail to capture highly context-dependent nuances of dialects, or (ii) rely on LLMs as translators (Gupta et al., 2024), which may have the very biases that our research wants to unveil (Fleisig et al., 2024; Smith et al., 2024). Therefore, we hire human AAVE speakers to rewrite instances of seven popular benchmarks to AAVE, including HumanEval (Chen et al., 2021), MBPP (Austin et al., 2021), and GSM8K (Cobbe et al., 2021) (see Section 2.1 for the complete list of datasets).

We build and release the first end-to-end human-written Standardized English-AAVE parallel benchmark called **ReDial** (Section 2, examples in Figure 1 and Appendix A.2). ReDial contains more than $1.2K$ Standardized English-AAVE prompt pairs, covering four fundamental reasoning tasks, namely **algorithm**, **math**, **logic**, and **comprehensive reasoning** (i.e., tasks requiring the composition of the other three reasoning skills). To the best of our knowledge, our dataset is **the first high-quality reasoning dataset with parallel prompts of Standardized English and a dialect annotated end-to-end by dialect speakers**. Unlike benchmarks using LLMs as judges, which are subjective to their internal biases (Zheng et al., 2023; Chen et al., 2024; Shi et al., 2024), ReDial offers an objective measure as judged by ground truth labels. It enables an easy, objective, and scalable way to report on the dialect fairness and robustness of LLMs as we keep the labels and evaluation process unchanged from the standard pipelines. We consider this dataset an important step toward revealing the robustness and fairness of state-of-the-art (SotA) LLMs for dialect users.

| Category | Algorithm (19.7%) | | Logic (29.8%) | | Math (25.8%) | | Comprehensive (24.7%) | Total |
|---|---|---|---|---|---|---|---|---|
| Source | HumanEval | MBPP | LogicBench | Folio | GSM8K | SVAMP | AsyncHow | - |
| Size | 164 | 150 | 200 | 162 | 150 | 150 | 240 | 1,216 |

Table 1: ReDial contains tasks for four categories, drawn from seven data sources. Percentage points in brackets for the categories indicate the proportion of corresponding data points in ReDial. In total, ReDial consists of $1,216$ fully-annotated parallel prompts.

We use ReDial to benchmark GPT-4o, GPT-4, LLaMA-3.1-70B-Instruct, and several other widely used SotA LLMs (Section 3). We discover that almost all LLMs suffer from significant performance drops for AAVE instances, despite the fact that they are semantically equivalent to their standardized counterparts. All models except GPT-4o and LLaMA-3.1-70B-Instruct have a pass rate of less than or similar to 0.6 in AAVE, even with Chain-of-thought prompting (CoT; Kojima et al., 2022; Wei et al., 2022), while the best pass rate in Standardized English is 0.832.

We further conduct an extensive analysis of the potential reasons for this performance gap (Section 4). We show that the skewness of dialect training data does not explain the whole picture, as large-scale LLMs have more difficulties in AAVE than misspelled Standardized English prompts, the latter of which LLMs are even less familiar with in the measurement of perplexity. This indicates that naively acquainting LLMs with AAVE by data augmentation might not be helpful for dialect robustness and fairness. Further, the performance gap cannot be easily closed by simple standardization: prompting LLMs to paraphrase AAVE in a standardized introduces higher costs, but cannot reach the Standardized English prompt performance. These findings point to the conclusion that LLMs are unfair and brittle to dialects and that the problem cannot be easily mitigated.

To summarise, the main contributions of this work are as follows:

1. We release ReDial, the first high-quality, human-annotated AAVE-Standardized English parallel dataset in four canonical reasoning tasks, comprising seven popular benchmarks.

2. We evaluate several SotA LLMs and show that they are significantly more brittle and unfair to AAVE prompts than their Standardized English counterparts, even with CoT.

3. Compared to misspelled Standardized English prompts of even higher perplexities, large-scale LLMs are more brittle to AAVE, which means that naive data augmentation might not solve the problem. We further find that prompting LLMs to rephrase a problem in Standardized English does not close the gap, either, but tends to introduce higher costs.

The the paper is organized as follows. We introduce ReDial in Section 2, and describe the benchmarking experiment and corresponding results in Section 3. We conduct extensive analysis in Section 4 and review related work in Section 5. Finally, we conclude the paper and discuss limitations, ethic statement, and reproducibility statement in Sections 6 to 8.

## 2 DATASET

In this section, we introduce **ReDial** (**Re**asoning with **Dial**ect Queries), a benchmark of more than $1.2K$ parallel Standardized English-AAVE query pairs (see a distribution overview in Table 1 and examples in Figure 1 and Appendix A.2). Following Zhu et al. (2023a), ReDial benchmarks four canonical reasoning tasks, namely **algorithm**, **logic**, **math**, and **comprehensive reasoning**. The task formulation is linguistically diverse, addresses cornerstone problems in human reasoning, and is of particular interest as it is challenging for LLMs.

In Section 2.1, we present more details about source data collection. In Section 2.2, we describe the annotation and validation process that we used to ensure that the data is of high quality.

### 2.1 DATA SOURCING

To obtain a highly curated dataset, we sample from seven widely used and established benchmarks. For each dataset, we report the key references, a description of the task, and the sample data size. We further provide example instances in Appendix A.1.

Algorithm **HumanEval** (Chen et al., 2021) contains 164 human-written instances as code completion tasks. We adopt the paradigm from InstructHumanEval[1] to convert code completion headings to instruction-following style natural language queries and include all of them in our benchmark.

Algorithm **MBPP** Austin et al. (2021) contains 1000 code generation queries. We include 150 randomly sampled data points from its sanitized test instances (Liu et al., 2023).

Math **GSM8K** (Cobbe et al., 2021) is a dataset of graduate school math questions written in natural language. It contains $8.79K$ instances in total. We randomly sample 150 instances from its test set.

Math **SVAMP** (Patel et al., 2021) contains 1000 instances of elementary-school math problems written in natural language. We randomly sample 150 instances from its test set.

Logic **LogicBench** (Parmar et al., 2024) is a benchmark of logic questions written in natural language. It contains logic questions of multi-choice and binary classification formats. We sample 100 instances from binary and multi-choice questions each, resulting in 200 instances in total.

Logic **Folio (original+perturbed)** (Han et al., 2022; Wu et al., 2023) Original Folio is a manually curated logic benchmark written by students in computer science in natural language. We select 81 instances with their manually perturbed versions from Wu et al. (2023), resulting in 162 instances in total.

Comprehensive **AsyncHow** (Lin et al., 2024) is a comprehensive reasoning benchmark in efficient planning with constraints. LLMs need to derive a dependency graph given natural language description (i.e., logic), find different possible paths in the graph (i.e., algorithm), and then calculate and compare the time needed for these paths (i.e., math) to reach the correct answer. We use this dataset to study whether LLMs' robustness is dependent on compositionality. We conduct stratified sampling according to the dataset's complexity metric and obtain 240 instances in total.

With data points from these sources, we construct a systematic reasoning benchmark with curated data. Then, we hire AAVE speakers to rewrite these data points in their dialect.

## 2.2 AAVE ANNOTATION AND QUALITY CHECK

We conduct a careful data annotation and quality check pipeline, which we schematize in Figure 2 and detail below.

**Annotation.** We hire AAVE speakers and instruct them to rewrite each instance by making them sound natural to them, but also preserve the essential information so that ground truth labels stay unchanged (e.g., it is allowed to turn "two" into 2, and vice-versa, but not to alter/delete numerical quantities). For algorithm tasks that require an understanding of code to keep the semantics, we specifically hire expert AAVE annotators with computer science backgrounds.[2]

**Validation.** To ensure the quality of the annotation, we conduct careful validations to ensure its **naturalness** and **correctness**. First, to ensure **naturalness**, we ask annotators to cross-check and edit each others' annotations to make sure that the annotations are natural to AAVE speakers. Second, to ensure **correctness**, we conduct both manual and automatic checks by non-AAVE speakers and LLMs. We first have non-AAVE speakers manually check whether the rewriting maintains the essential information and send the invalid instances back to AAVE speakers for reannotation. We conduct a sanity check with GPT-4o for the correctness of rewriting (details in Appendix A.4). We **manually check** data that GPT-4o flags as invalid to see if all essential information is preserved: we stress that in this round **no instance is rejected solely based on the LLM's judgment**. We return invalid instances to AAVE speakers for correction and iterate the process until all the data passes the check.

After this process, we obtain a high-quality, human-annotated dataset ReDial with more than $1.2K$ Standard English-AAVE parallel prompts in four canonical reasoning tasks. ReDial is the first benchmark of its kind and enables easy testing and analysis of LLMs' dialect fairness and robustness

---

[1] https://huggingface.co/datasets/codeparrot/instructhumaneval
[2] Please refer to Appendix A.3 for annotators' compensation, qualification, and other guideline details.

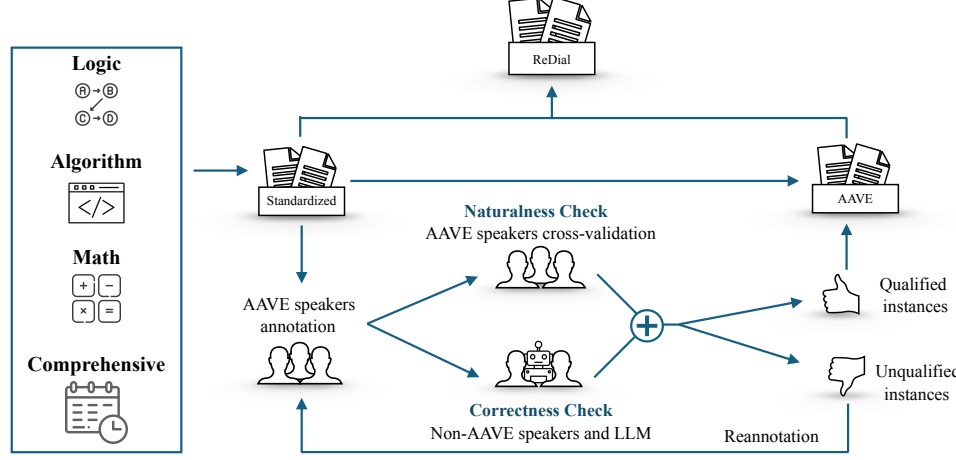

Figure 2: Annotation and cross-validation of ReDial instances. We first sample instances from datasets of four canonical reasoning tasks to compose the source data, then we hire AAVE speakers to rewrite the instances in their dialect. To ensure the high quality of the rewritten data, we conduct **naturalness check** by AAVE speakers and **correctness check** by non-AAVE speakers and LLMs. We reannotate instances that do not pass the quality checks and iterate the process until the data meet our criteria. Finally, we combine the source data and AAVE rewriting to obtain a high-quality parallel reasoning dataset ReDial.

in reasoning tasks. In the rest of the paper, we will refer to the Standardized English part of ReDial as *Standardized ReDial*, and its AAVE part as *AAVE ReDial*.

## 3 EXPERIMENT

In this section, we benchmark several SotA LLMs on the parallel prompts from ReDial. We report experiment setting in Section 3.1 and results in Section 3.2.

### 3.1 EXPERIMENTAL SETTING

We test four families of models, one proprietary and three open-source, on zero-shot prompting and zero-shot Chain of Thought (Kojima et al., 2022; Wei et al., 2022) to simulate one setting for general users and one setting for expert users. We deliberately do not test more advanced prompting methods such as Tree of Thought (Yao et al., 2024) and Self-Refine (Madaan et al., 2024) as **we are interested in how LLMs behave when prompted for daily usage** since this is the context in which input is most likely to contain dialectal features.

We elaborate further on model choices in Section 3.1.1 and experiment settings in Section 3.1.2.

### 3.1.1 MODELS

Here, we report the details about the models we test. The rationale is to benchmark widely used LLMs with impressive reasoning performance.

**GPT.** We use GPT-4o, GPT-4, GPT-3.5-turbo (Achiam et al., 2023),[3] as a family of SotA closed-source models to compare with open-source models for dialect robustness.

**LLaMA.** We use LLaMA-3-8B/70B-Instruct and LLaMA-3.1-70B-instruct (Dubey et al., 2024) which are reported for comparable performance with proprietary GPT models.

---

[3]https://openai.com/index/hello-gpt-4o, https://openai.com/index/gpt-4/, https://platform.openai.com/docs/models/gpt-3-5-turbo.

| Model | Setting | Original | AAVE |
|-------|---------|----------|------|
| GPT-4o | Zero-shot | 0.832 | $\mathbf{0.716}_{\triangle=0.116}$ |
| | CoT | 0.826 | $\mathbf{0.784}_{\triangle=0.043}$ |
| GPT-4 | Zero-shot | 0.678 | $\mathbf{0.612}_{\triangle=0.067}$ |
| | CoT | 0.706 | $\mathbf{0.590}_{\triangle=0.115}$ |
| GPT-3.5-turbo | Zero-shot | 0.531 | $\mathbf{0.460}_{\triangle=0.072}$ |
| | CoT | 0.517 | $\mathbf{0.416}_{\triangle=0.101}$ |
| LLaMA-3.1-70B-Instruct | Zero-shot | 0.663 | $\mathbf{0.599}_{\triangle=0.064}$ |
| | CoT | 0.759 | $\mathbf{0.711}_{\triangle=0.049}$ |
| LLaMA-3-70B-Instruct | Zero-shot | 0.628 | $\mathbf{0.562}_{\triangle=0.066}$ |
| | CoT | 0.693 | $\mathbf{0.622}_{\triangle=0.072}$ |
| LLaMA-3-8B-Instruct | Zero-shot | 0.489 | $0.480_{\triangle=0.009}$ |
| | CoT | 0.488 | $0.472_{\triangle=0.016}$ |
| Mixtral-8x7B-Instruct-v0.1 | Zero-shot | 0.388 | $\mathbf{0.274}_{\triangle=0.114}$ |
| | CoT | 0.431 | $\mathbf{0.345}_{\triangle=0.086}$ |
| Mistral-7B-Instruct-v0.3 | Zero-shot | 0.297 | $\mathbf{0.214}_{\triangle=0.083}$ |
| | CoT | 0.305 | $\mathbf{0.252}_{\triangle=0.053}$ |
| Phi-3-Medium-128K-Instruct | Zero-shot | 0.513 | $\mathbf{0.454}_{\triangle=0.059}$ |
| | CoT | 0.513 | $\mathbf{0.458}_{\triangle=0.055}$ |
| Phi-3-Small-128K-Instruct | Zero-shot | 0.530 | $\mathbf{0.421}_{\triangle=0.109}$ |
| | CoT | 0.549 | $\mathbf{0.429}_{\triangle=0.119}$ |
| Phi-3-Mini-128K-Instruct | Zero-shot | 0.456 | $\mathbf{0.410}_{\triangle=0.046}$ |
| | CoT | 0.528 | $\mathbf{0.461}_{\triangle=0.067}$ |

Table 2: Pass rates for testing models with zero-shot and CoT prompting on ReDial. We follow the recommendations from Dror et al. (2018) and test the statistical significance of performance differences between Standardized English and AAVE using the McNemar's test for binary data (McNemar, 1947). We correct p-values for multiple measurements using the Holm-Bonferroni method (Holm, 1979). Results in **bold** show a statistically significant deviation between AAVE and Standardized ReDial (i.e., models have significant drops in AAVE). We also indicate the absolute delta in performance between the two settings.

**Mistral/Mixtral.** We use Mistral-7B-Instruct-v0.3 (Jiang et al., 2023) and Mixtral-8x7B-Instruct-v0.1 (Jiang et al., 2024). Mistral-7B-Instruct-v0.3 is reported to be outstanding in reasoning; with Mixtral-8x7B-Instruct-v0.1, we can understand whether Mixture-of-Expert architectures enhance dialect robustness.

**Phi.** We use Phi-3-Mini/Small/Medium-128K-Instruct (Abdin et al., 2024; Gunasekar et al., 2023) in our experiment. Phi-3 models, trained on carefully designed "textbook" data, are reported for impressive performance in reasoning despite their small sizes (3.8/7/14B parameters each). We use these models to understand how (i) scaling laws (Kaplan et al., 2020) and (ii) highly curated training data affect LLMs' dialect robustness and fairness.

### 3.1.2 IMPLEMENTATION AND EVALUATION

**Implementation.** We use temperature zero for all experiments to ensure maximum reproducibility. We report two prompting methods in our main results: (i) *zero-shot* (i.e., directly prompting LLMs with task instances, which resembles general real-life use cases the most) and (ii) zero-shot Chain of Thought (Wei et al., 2022; Kojima et al., 2022) (CoT, i.e., adding instructions in the spirit of "`Let's think step by step`" on top of task descriptions, which resembles expert user prompts to improve model performance).[4] We report further implementation details in Appendix A.5.

**Evaluation.** To unify evaluation metrics, we consider the pass rate for all tasks. For Algorithm, we consider Pass@1 using all base and extra unit test cases in EvalPlus (Liu et al., 2023), which results

---

[4]We also test non-zero temperatures and report results in Appendix A.6.

|  |  | Algorithm | Math | Logic | Comprehensive | Average |
|---|---|---|---|---|---|---|
| Zero-shot | Original | 0.602 | 0.733 | 0.578 | 0.191 | 0.546 |
|  | AAVE | $\mathbf{0.517}_{\Delta=0.085}$ | $\mathbf{0.665}_{\Delta=0.068}$ | $\mathbf{0.522}_{\Delta=0.056}$ | $\mathbf{0.101}_{\Delta=0.090}$ | $\mathbf{0.473}_{\Delta=0.073}$ |
| CoT | Original | 0.597 | 0.811 | 0.580 | 0.240 | 0.574 |
|  | AAVE | $\mathbf{0.495}_{\Delta=0.102}$ | $\mathbf{0.742}_{\Delta=0.068}$ | $\mathbf{0.530}_{\Delta=0.050}$ | $\mathbf{0.177}_{\Delta=0.063}$ | $\mathbf{0.504}_{\Delta=0.070}$ |

Table 3: Pass rates by task averaged across responses from all models with zero-shot and CoT prompting. Results in **bold** show a statistically significant deviation according to McNemar's tests applied to AAVE and Standardized English (i.e., models have significant drops in AAVE). We also indicate the absolute delta in performance between the two settings.

in either pass or fail for every code generation. We convert all other task measures of correctness or incorrectness to pass or fail.

## 3.2 EXPERIMENTAL RESULTS

We report pass rates for ReDial in Table 2 and 3, respectively averaged by task and by model (see detailed results in Appendix A.7). We now summarise the main results of our experiments.

**All models are brittle to AAVE.**  We find that all models experience performance drops in AAVE compared to Standardized ReDial, and these drops are statistically significant in all cases, with the sole exception of LLaMA-3-8B-Instruct. Except for GPT-4o and LLaMA-3.1-70B-Instruct, **all other models have pass rates of similar to or below 0.6 in AAVE Redial, even with CoT**, while the best pass rate in Standardized ReDial is 0.832. This indicates that our benchmark poses huge challenges to models, both in terms of absolute performance and with respect to their dialect robustness and fairness.

**All reasoning tasks are brittle to AAVE.**  LLMs experience the most severe drops in tasks related to algorithm and comprehensive reasoning. In comprehensive tasks that require the composition of more than one elementary reasoning skill, the relative performance drop is especially strong: almost $50\%$ relative performance drop across models with zero-shot, and close to $30\%$ drop with CoT. LLMs face further difficulty when they are asked in a dialect to compose different skills for solving problems.

**Scaling does not make models more robust to AAVE.**  Comparing within LLaMA-3 and Phi-3 model familes, we find that although increasing model size improves absolute performance, it cannot close the Standardized English-AAVE performance gaps. For example, comparing LLaMA-3-8B with LLaMA-3-70B, the performance gap in zero-shot widens from 0.009 to 0.066 (Table 2) Thus, scaling does not always result in better dialect robustness and fairness. We also find that Mixtral-8x7B-Instruct-v0.1 has an even bigger drop compared to the smaller Mistral-7B-Instruct-v0.3. This suggests that Mixture-of-Experts does not necessarily bring performance gains to models prompted in dialects either.

**Highly curated data is particularly brittle to AAVE.**  The Phi-3 models, which are trained on highly curated clean data, achieve impressive performance despite their small sizes in Standardized ReDial. For instance, Phi-3-Mini-128K-Instruct (3.8B) outperforms LLaMA-3-8B-Instruct (8B) in the Standardized ReDial with CoT prompting (0.528 vs. 0.488 pass rate). However, it suffers from a large (0.067) performance drop in AAVE ReDial, while LLaMA only drops by less than 0.016, a result that is not statistically significant. This finding is in line with Dodge et al. (2021), which suggests that cleaning and removing data exacerbates unfairness to minority groups.

## 4 ANALYSIS OF BRITTLENESS TO AAVE

This section investigates the links between dialectal features and the AAVE brittleness. We compare model performance on human-written AAVE data and misspelled English inputs (Section 4.1) to show that AAVE training data skewness does not explain the whole picture of dialect unfairness

|  | LLaMA-3.1-70B-Instruct | Phi-3-Medium-128K-Instruct | Phi-3-Mini-128K-Instruct |
|---|---|---|---|
| Standardized | 9.4 | 5.9 | 7.1 |
| AAVE | 17.5 | 12.5 | 15.9 |

Table 4: Averaged perplexities across instances calculated by different models on Standardized/AAVE ReDial.

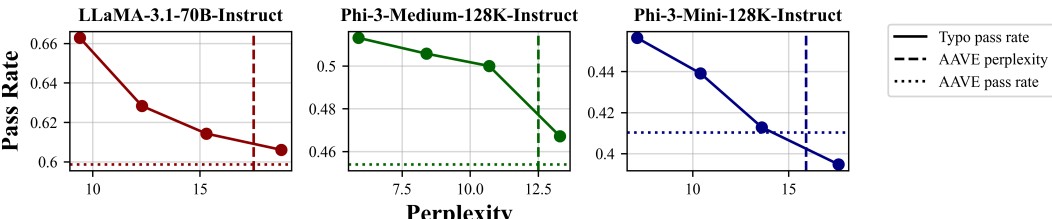

Figure 3: Model performance on misspelled Standardized English compared to human-written AAVE data. We gradually add noise to Standardized ReDial to increase its perplexities until they surpass the perplexity of AAVE ReDial and report the models' performance on every perturbation level. Horizontal and vertical lines refer to model pass rates/perplexities on AAVE ReDial respectively. Larger LLMs (i.e., LLaMA-3.1-70B-Instruct and Phi-3-Medium-128K-Instruct) perform worse on AAVE than on perturbed text with a similar perplexity level.

and brittleness. We further show that asking a model to rephrase an AAVE input into Standardized English and then answer the question does not cancel the unfairness but tends to increase the computational cost (in terms of tokens generated, Section 4.2). Last, we qualitatively examine cases where LLMs fail in AAVE, even after rephrasing in Standardized English, but succeed in the prompts that are originally written in Standardized English, and identify key error patterns for them (Section 4.3).

## 4.1 Data Skewness Does not Explain AAVE Brittleness

One possible explanation of the performance drop on AAVE is its infrequency in LLMs' training corpora. As the model's training data is largely unknown, we use perplexity as a proxy to measure how familiar the LLMs are with some data: the higher the perplexity, the less familiar an LLM is with the data. We conduct experiments on LLaMA-3.1-70B-Instruct, Phi-3-Medium/Mini-128K-Instruct on Standardized and AAVE ReDial and report their perplexities averaged across instances in Table 4.

As expected, LLMs have higher perplexities on AAVE than Standardized ReDial, which indicates they are indeed less familiar with AAVE than with Standardized English. Does this mean that we can fully attribute the dialect performance gap to its data skewness? To answer this question, we gradually perturb Standardized English by injecting typos, such that we decrease the LLMs' familiarity with the input texts (i.e., the measured perplexity goes up). Specifically, we simulate typos by replacing/deleting/adding characters in Standardized ReDial. We control an increasing perturbation rate until the tested models' perplexities exceed those measured in AAVE (i.e., when models are less familiar with misspelled Standardized English than with AAVE).[5]

Results are in Figure 3. Interestingly, although LLaMA-3.1-70B-Instruct and Phi-3-Medium-128K-Instruct performance drops with denser perturbations, even their drop in the strongest perturbation level is lower than that of human-written dialect prompts. **This means that even when these LLMs are more familiar with AAVE, they still cannot perform as well in this dialect.** Conversely, we find that Phi-3-Mini-128K-Instruct has better performance in AAVE data compared to perturbed texts of similar perplexities. This discrepancy seems to suggest that the small-scale model might have a different behavior pattern compared to larger models in dialect robustness. We further find that the denser AAVE features are, the bigger the performance drop is. We report further details in Appendix A.8, where we gradually control and inject synthetic lexico-syntactic dialect features following Ziems et al. (2022).

---

[5]In practice, we introduce perturbations of densities $\{0, 0.02, 0.04, 0.06\}$, which results in four different typo perplexity levels for each model.

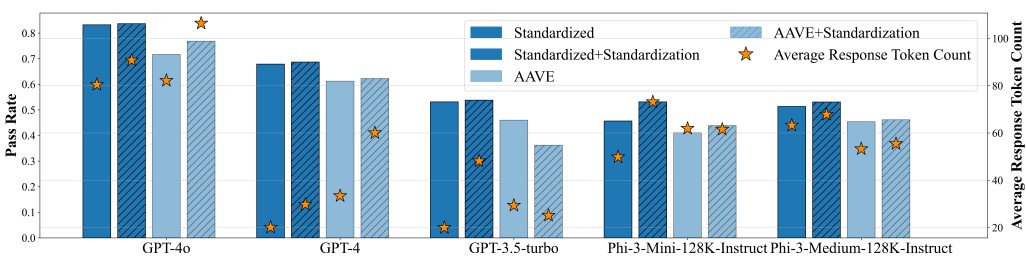

Figure 4: Model pass rate and average response token count before and after being prompted for standardization. Standardization prompting generally improves LLM performance in both Standardized and AAVE ReDial (bar plot). However, even AAVE ReDial with standardization prompting cannot reach LLMs' vanilla performance in Standardized ReDial, despite that they also tend to result in more tokens generated (scatter plot).

Generally, the findings in this subsection suggest that (i) the unfamiliarity of LLMs to AAVE does not explain the whole picture of the performance drop, so naively increasing AAVE in the training data may not diminish the performance gap, and (ii) LLMs, especially at large scales, might be even more brittle when facing the language of real users than what has been suggested by the previous robustness literature based on typo-style prompts (Zhu et al., 2023b).

## 4.2 Rephrasing Prompts in Standardized English Does Not Fill the AAVE Gap

Since LLMs generally show superior performance in Standardized ReDial, we experiment with instructing models to standardize and then answer the question to mitigate the AAVE bias, which we refer to as *standardization*. Specifically, we suffix *'Let's rephrase the query in Standard English first, then answer the question'* to every query. Results are reported in Figure 4 (bar plot).

Indeed, LLM performance generally increases with standardization. Surprisingly, standardization improves model performance even when the prompt input is already in Standard English. Despite this, **their performance on AAVE ReDial with standardization promoting still cannot reach their vanilla performance on Standardized ReDial**. We further analyze the error patterns in Section 4.3.

Moreover, we notice that standardization introduces a computational overhead in terms of token count of LLMs' responses (Figure 4, scatter plot), especially in GPT-4o and GPT-4. **This means that even if dialect users pay more, they might still not be able to receive the same quality service as users who use Standardized English.**

## 4.3 Qualitative Analysis

Intuitively, standardization prompting should cancel the dialect gap. However, we still observe a sensible gap between model performance on Standardized ReDial with zero-shot prompting and AAVE ReDial with standardization prompting. In this section, we qualitatively compare GPT-4o's outputs in these two settings, the model with the best absolute overall performance, and examine its errors. We focus on the math subset of ReDial and identify three key error patterns: **wrong question rephrasing**, **distraction by irrelevant information**, and **failure to execute all steps**.

**Wrong question rephrasing.** GPT-4o wrongly phrases question '*Jame ... How many years have they got between them now if in 8 years his cousin will be 5 years younger than twice his age?*' to '*James ... How old is his cousin now?*', which changes the question of age gap to absolute age.

**Distraction by irrelevant information.** GPT-4o gets distracted by task-irrelevant information after AAVE standardization while the distraction is not observed in Standardized ReDial. For instance, in '*Say we got 8 different books and 10 different movies in the crazy silly school series. How many more movies than books is there gon be in the crazy silly school series if you read 19 books and watched 61 movies?*', books that have been read and movies that have been watched are not associated with the answer. Although GPT-4o can ignore irrelevant information in Standardized ReDial, it gets distracted after AAVE standardization, which shows the brittleness of its reasoning ability.

**Failure to execute all the steps.** GPT-4o sometimes simulates an algorithm to solve math problems after standardization (e.g., '*Let $(x)$ be the number of apple pie boxes...*'). However, it does not fully solve the problem in the end and only returns a formula (e.g., '*30x + 255*'), which indicates that the model's reasoning ability is limited when it comes to program simulation for queries expressed in dialects.

## 5 RELATED WORKS

**Dialect studies in natural language processing.** Previous works on AAVE studies in natural language processing mostly focus on non-reasoning-heavy tasks such as POS tagging (Jørgensen et al., 2015; 2016), language identification and dependency parsing (Blodgett et al., 2016), automatic captioning (Tatman, 2017), and general language generation (Deas et al., 2023). AAVE is also found to be more likely to trigger false positives in hate speech identifiers (Davidson et al., 2019; Sap et al., 2019) due to specific word choices (Harris et al., 2022), be considered negative by automatic sentiment classifier (Groenwold et al., 2020), and cause covert biases in essential areas of social justice (Hofmann et al., 2024). Relevant studies (Ziems et al., 2022; Gupta et al., 2024) also find that rule-based AAVE feature perturbations can downgrade language model performance in a wide range of tasks covered by GLUE (Wang, 2018).

More generally, dialects in world languages pose challenges to natural language processing systems. Ziems et al. (2023) find that auto-encoder models are brittle on rule-based English dialect feature perturbations. Fleisig et al. (2024) report that English dialect speakers perceive responses generated by chatbots to be more negative than Standardized English prompts. Faisal et al. (2024) find that world dialects cause problems in tasks including dependency parsing (Scherrer et al., 2019) and machine translation (Mirzakhalov, 2021) on mBERT and XLM-R (Conneau et al., 2020).

However, existing works fail to systematically cover reasoning tasks in dialects. There is no existing high-quality end-to-end human-annotated dataset on such a task. Moreover, studies on LLM task-specific capabilities tend to focus on traditional auto-encoder models such as BERT and RoBERTa instead of SotA auto-regressive LLMs. Our work fills the gaps in these areas.

**Fairness and Robustness of Large Language Models**. LLMs are widely testified to be both unfair and brittle. They introduce unfair performance (Huang et al., 2023; Dong et al., 2024) and cost (Petrov et al., 2024) to users across different languages, exacerbate social imbalance by marginalizing minority groups in various aspects including gender (Kotek et al., 2023; Fraser & Kiritchenko, 2024), race (Hofmann et al., 2024; Wang et al., 2024), and culture (Naous et al., 2023; Tao et al., 2024). Our work shows for the first time that LLMs also exhibit unfairness in reasoning tasks for speakers of a dialect.

Previous works report that LLMs are very brittle to slight variations of prompts by introducing typos or paraphrasing in Standardized English (Elazar et al., 2021; Liang et al., 2022; Raj et al., 2022; Zhu et al., 2023b; Lin et al., 2024). In this work, we consider a novel application of using human-written perturbations in AAVE by asking humans to rewrite instances to their dialect and evaluate LLM robustness towards these natural perturbations, which have proven to cause LLMs to be more brittle than synthetic typo-style (Section 4.1) or linguistic-rule-based (Appendix A.8) perturbations.

## 6 CONCLUSION

In this work, we present ReDial which has $1.2K+$ parallel Standardized English-AAVE prompts to evaluate LLMs' dialect robustness and fairness in algorithm, logic, math, and comprehensive reasoning as four canonical reasoning tasks. With ReDial, we find that SotA LLMs show significant unfairness and brittleness to reasoning tasks expressed in AAVE. The data skewness of AAVE does not explain the whole picture as large-scale LLMs are more brittle to AAVE compared to Standardized English typos of even higher perplexities. Prompting LLMs to rephrase questions in Standardized English cannot fully bridge the gap but tends to introduce higher costs. These findings highlight the unfairness of LLMs to dialect users and also shed light on the brittleness of LLMs' reasoning capabilities when it comes to minor variations of prompts without changing their semantics. We call for further studies to enhance LLMs' fairness and robustness to dialects to provide equal service to users from all linguistic groups and demographics.

## 7 ETHICS STATEMENT

ReDial is a collection of high-quality human-annotated translations: obtaining such data requires making clear design choices and poses ethical questions that we hereby address.

For data collection, we deliberately do not set hard constraints for annotator identity and demographic verification, recognizing there are no definite boundaries to identify dialects and their speakers (King, 2020). King (2020) further elaborate that the term "AAVE" itself is contested, with alternatives that could be used instead; in employing the term "AAVE", we adhere to the widely used terminology in related works on dialects and NLP (Ziems et al., 2022; Gupta et al., 2024). We corroborate the data quality by asking self-identified dialect speakers to cross-validate each others' answers.

We do not collect annotators' personal information; while we firmly commit to this rule to protect annotators' privacy, it makes it difficult to draw conclusions about how annotators' backgrounds shape their writing/individual-level variations. Further on the ethical aspect of data collection, we work with a data vendor that makes sure the recruitment and annotation adhere to high standards for and from the annotators. However, although we have a legal contract and we try our best to convey our guidelines and requirements, we admit that we do not have full control over how the vendor recruits people and conducts data annotation.

We also stress that the LLM validation stage in our quality control process is not completely trustworthy as even they are prone to hallucinations (Ji et al., 2023) and biases against minority groups (Xu et al., 2021; Fleisig et al., 2024; Smith et al., 2024; Wang et al., 2024). To mitigate this issue, we conduct full manual checks of every instance identified as invalid by an LLM so that no instance is rejected purely because of LLM decisions.

Last, there are limitations on how well standard benchmarks reflect use cases of practical usage for LLMs. For ReDial, we select the source datasets among those reported in highly impactful LLM technical reports such as GPT-4 (Achiam et al., 2023), LLaMA-3 (Dubey et al., 2024), and Phi-3 (Abdin et al., 2024). Their popularity makes it easy to integrate them with existing pipelines, and the presence of ground truth labels mitigates inherent biases of using LLMs as evaluators (Zheng et al., 2023; Chen et al., 2024; Shi et al., 2024). Although we try our best to simulate user queries (e.g., changing code completion to instruction following queries in HumanEval), we do note there can be a gap between tasks as in standard benchmarks and queries in real workflows.

## 8 REPRODUCIBILITY STATEMENT

Code can be found at `https://anonymous.4open.science/r/redial_eval-0A88` and the dataset will be released upon publication.

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

# A APPENDIX

## A.1 SOURCE DATASET ILLUSTRATION

### A.1.1 ALGORITHM

---

**Original HumanEval**

```python
from typing import List

def has_close_elements(numbers: List[float], threshold: float)
-> bool:
    """ Check if in given list of numbers, are any two numbers
    closer to each other than given threshold.
    >>> has_close_elements([1.0, 2.0, 3.0], 0.5)
    False
    >>> has_close_elements([1.0, 2.8, 3.0, 4.0, 5.0, 2.0], 0.3)
    True
    """
```

**InstructHumanEval Used in the Paper**

Write a function has_close_elements(numbers: List[float], threshold: float) -> bool to solve the following problem:
Check if in given list of numbers, are any two numbers closer to each other than given threshold.
>>> has_close_elements([1.0, 2.0, 3.0], 0.5)
False
>>> has_close_elements([1.0, 2.8, 3.0, 4.0, 5.0, 2.0], 0.3)
True

---

**MBPP**

Write a python function to remove first and last occurrence of a given character from the string.
Your code should pass these tests:
assert remove_Occ("hello","l") == "heo"
assert remove_Occ("abcda","a") == "bcd"
assert remove_Occ("PHP","P") == "H"

---

### A.1.2 LOGIC

---

**LogicBench**

If an individual consumes a significant amount of water, they will experience a state of hydration. Conversely, if excessive amounts of sugar are ingested, a sugar crash will ensue. It is known that at least one of the following statements is true: either the Jane consumes ample water or she will not experience a sugar crash. However, the actual veracity of either statement remains ambiguous, as it could be the case that only the first statement is true, only the second statement is true, or both statements are true.
Can we say at least one of the following must always be true? (a) she will feel hydrated and (b) she doesn't eat too much sugar

---

**Folio**

Consider the following premises: "People in this club who perform in school talent shows often attend and are very engaged with school events. People in this club either perform in school talent shows often or are inactive and disinterested community members. People in this club who chaperone high school dances are not students who attend the school. All people in this club who are inactive and disinterested members of their community chaperone high school dances. All young children and teenagers in this club who wish to further their academic careers and educational opportunities are students who attend the school. Bonnie is in this club and she either both attends and is very engaged with school events and is a student who attends the school or is not someone who both attends and is very engaged with school events and is not a student who attends the school."

Assuming no other commonsense or world knowledge, is the sentence "Bonnie performs in school talent shows often." necessarily true, necessarily false, or neither? Answer either "necessarily true", "necessarily false", or "neither".

### A.1.3  MATH

**GSM8K**

Given a mathematics problem, determine the answer. Simplify your answer as much as possible and encode the final answer in <answer></answer> (e.g., <answer>1</answer>). Question: Janet's ducks lay 16 eggs per day. She eats three for breakfast every morning and bakes muffins for her friends every day with four. She sells the remainder at the farmers' market daily for $2 per fresh duck egg. How much in dollars does she make every day at the farmers' market?
Answer:

**SVAMP**

Given a mathematics problem, determine the answer. Simplify your answer as much as possible and encode the final answer in <answer></answer> (e.g., <answer>1</answer>). Question: Winter is almost here and most animals are migrating to warmer countries. There are 41 bird families living near the mountain. If 35 bird families flew away to asia and 62 bird families flew away to africa How many more bird families flew away to africa than those that flew away to asia?
Answer:

### A.1.4 COMPREHENSIVE

---

**AsyncHow**

To create a video game, here are the steps and the times needed for each step.
Step 1. Learn the basics of programming (180 days)
Step 2. Learn to use a language that is used in games (60 days)
Step 3. Learn to use an existing game engine (30 days)
Step 4. Program the game (90 days)
Step 5. Test the game (30 days)

These ordering constraints need to be obeyed when executing above steps:
Before starting step 2, complete step 1.
Before starting step 3, complete step 1.
Before starting step 4, complete step 2.
Before starting step 4, complete step 3.
Before starting step 5, complete step 4.

Question: Assume that you need to execute all the steps to complete the task and that infinite resources are available. What is the shortest possible time to create a video game? Answer the time in double quotes.
Answer:

---

## A.2  ReDial Samples

---

**Algorithm**

Standardized

Write a function python_function(numbers: List[float], threshold: float) $->$ bool to realize the following functionality:
Check if in given list of numbers, are any two numbers closer to each other than given threshold.
$>>>$ python_function([1.0, 2.0, 3.0], 0.5)
False
$>>>$ python_function([1.0, 2.8, 3.0, 4.0, 5.0, 2.0], 0.3)
True
Generate a Python function to solve this problem. Ensure the generated function is named as python_function.

AAVE

Aight, so here you gonna write a function called python_function(numbers: List[float], threshold: float) $->$ bool that gon' do this following functionality:
Aight, Listen. Say you got a list of numbers yeah? Now, we trynna see if any two of 'em numbers is closer to each other than a number you give, feel me?So, this is what we 'bout to do:
$>>>$ python_function([1.0, 2.0, 3.0], 0.5)
False
That's gon' give you False cuz ain't none of 'em numbers close enough.But, if you hit it like:
$>>>$ python_function([1.0, 2.8, 3.0, 4.0, 5.0, 2.0], 0.3)
True
Bet you gettin' True, cuz this time some of 'em numbers real tight.
You gotta whip up a Python function to handle this problem. You gon' make sure the function name right, which gotta python_function.

---

**Math**

Standardized

Given a mathematics problem, determine the answer. Simplify your answer as much as possible and encode the final answer in $< answer >< /answer >$ (e.g., $< answer > 1 < /answer >$).
Question: John is raising money for a school trip. He has applied for help from the school, which has decided to cover half the cost of the trip. How much money is John missing if he has \$50 and the trip costs \$300?
Answer:

AAVE

"Bet, so here's whatsup. Youn finna get a math problem, and you gon' tryna find the answer out. You gotta simplify that answer as much as possible tehn wrap it up inside $< answer >< /answer >$ (somethin' like this:, $< answer > 1 < /answer >$).
Question: John been raisin' money fo' a school trip. He done ask the school fo' help, and they decided they gon' be coverin' half the trip cost. How much money John be missin' if he got \$50, and the trip cost \$300.
Answer:

**Logic**

Standardized

Consider the following premises: "All bears in zoos are not wild.
Some bears are in zoos. "
Assuming no other commonsense or world knowledge, is the sentence "Not all bears are wild." necessarily true, necessarily false, or neither? Answer either "necessarily true", "necessarily false", or "neither". Encode the final answer in $< answer >< /answer >$ (e.g., $< answer >$necessarily true$< /answer >$).

AAVE

Aight, check this. You got 'em premises right here: "All bears in zoos ain't considered wild. There are some bears livin' in zoos. "
Ain't no using no other commonsense or world knowledge, you gon' try find out if the sentence "Not every bear out there be wild." necessarily true, necessarily false, or neither? Pick either "necessarily true", "necessarily false", or "neither". Then wrap that answer up in $< answer >< /answer >$ (e.g., $< answer >$necessarily true$< /answer >$).

**Comprehensive**

Standardized

To try fishing for the first time, here are the steps and the times needed for each step
Step 1. drive to the outdoor store (10 minutes)
Step 2.compare fishing poles (30 minutes)
Step 3. buy a fishing pole (5 minutes)
Step 4. buy some bait (5 minutes)
Step 5. drive to a lake (20 minutes)
Step 6. rent a small boat (15 minutes)

These ordering constraints need to be obeyed when executing above steps:
Step 1 must precede step 2.
Step 2 must precede step 3.
Step 2 must precede step 4.
Step 3 must precede step 5.
Step 4 must precede step 5
Step 5 must precede step 6.

Question: Assume that you need to execute all the steps to complete the task and that infinite resources are available. What is the shortest possible time to complete this task? What is the shortest possible time to complete this task? Encode the final answer in $< answer ><$ $/answer >$ (e.g., $< answer >$1 min$< /answer >$).
Answer:

AAVE

If you finna go fish for the first time, here's what you got to know and the times you need for each step.
Step 1. To kick things off, pull up to the outdoor store (10 minutes)
Step 2. Check out which one of them fishing poles is good and which one is not (30 minutes)
Step 3. Cop a fishing pole (5 minutes)
Step 4.Get yourself some bait as well (5 minutes)
Step 5. Head out to a lake (20 minutes)
Step 6.rent yourself a small boat (15 minutes)

These ordering constraints gotta be followed when you doin' 'em steps above: You gotta deal with 1 before hittin' the 2.
You gotta deal with 2 before hittin' the 3.
You gotta deal with 2 before hittin' the 4.
You gotta deal with 3 before hittin' the 5.
You gotta deal with 4 before hittin' the 5.
You gotta deal with 5 before hittin' the 6.

Question: Assumin' you outta do all 'em steps to finish up the task, and you got infinite resources. What the shortest time be to knock this task out? Wrap that answer up in $< answer ><$ $/answer >$ (e.g., $< answer >$1 min$< /answer >$).
Answer:

## A.3 RUBRICS

### A.3.1 EMPLOYMENT INFORMATION

We work with data vendors to employ 13 annotators in total for our task. For algorithm instance annotation, we specifically hire annotators with computer science backgrounds. Annotators are self-identified as proficient speakers of African American Vernacular English. We do not pose any hard constraints in verifying dialect identity as previous studies do (e.g., Ziems et al. (2023)). We note even within a dialect there can be significant variations on the individual level and that we want to avoid homogenization and over-simplification of the dialect (King, 2020). Instead, we ask self-identified annotators to cross-check each other's annotations and modify if they sound unnatural.

Details of employment are shown below.

**Information Collected** We do not collect personally identifiable information from our annotators (e.g., name, age, etc). We only collect the annotators' responses to our consent form and their annotations of our data.

**Risk and Consent** We note that our base datasets are from publicly available, widely used, peer-reviewed datasets that adhere to peer-review regulations. Moreover, our tasks are mainly centered around reasoning, which does not concern sensitive information per se. In addition, we make sure that annotators understand the risks of the annotation (i.e., although we have tried our best to ensure the safety of the data, it is still possible that they may feel uncomfortable in the annotation) and their right to exit the task during the process by signing a consent form prior to the start of the task.

**Compensation** We offer payment to annotators with hourly rates higher than the U.S. federal minimum wage.

**No AI Assistant** We explicitly inform our annotators that they should not reply on any AI assistant tools to help them complete the task. To further ensure this, we design our annotation platform to disallow copy and paste. The default annotation area for annotators is the original text, which means that it is easier for annotators to simply edit the text than querying AI assistants.

### A.3.2 ANNOTATION GUIDELINE

You need to translate/rephrase/localize the task input in a way that is natural to the speakers of your dialect without changing the intention of the prompts. You should not change named entities, numbers, equations, variable names and other formal devices that are not natural language per se or those that would affect the intention of the prompts. The translation does not need to be grammatical or acceptable in standard English. Rather, it should accurately reflect the features of their dialects. You can add or delete some functional content to make the prompts sound more natural (e.g., adding fillers). However, you should keep the vital information complete and unchanged.

You should NOT change information that would invalidate the output given the question. If you are unsure about any specific parts, leave them unchanged. Especially, you should not change the following parts:

(i) numbers (e.g. 180 in 180 days)

(ii) units (e.g. days in 180 days)

(iii) equations and symbols (e.g., \[f(x) = \left \{ \begin{array}{cl} ax+3, & \text{ if }$x > 2$ in Let \[$f(x) = \left \{ \begin{array}{cl} ax + 3, & \text{ if }x > 2$)

(iv) proper nouns (e.g., Natalia in Natalia sold clips to 48 of her friends)

(v) function names, variables, data types, and input-output examples (e.g., $>>>$ has_close_elements($[1.0, 2.0, 3.0]$, 0.5) False $>>>$ has_close_elements($[1.0, 2.8, 3.0, 4.0, 5.0, 2.0]$, 0.3) True in Check if in given list of numbers, are any two numbers closer to each other than given threshold. $>>>$ has_close_elements($[1.0, 2.0, 3.0]$, 0.5) False $>>>$ has_close_elements($[1.0, 2.8, 3.0, 4.0, 5.0, 2.0]$, 0.3) True)

## A.4 DATA QUALITY VERIFICATION

After we conduct human validations for *naturalness* and *correctness* of prompts, we conduct the final round sanity check with GPT-4o. We prompt GPT-4o with temperature 0.7 and sample three instances for each query. We manually inspect instances again where all of the answers suggest that they are invalid paraphrases of the original prompts.

---

**User prompt**

You will be given two prompts, one in Standard English and one in African American English. Determine whether the African American English prompt is a valid paraphrase of the Standard English prompt. Ignore the semantic validaty of the Standard English prompt.
Standard English: "[SAE_PROMPT]"
African American English: "[AAVE_PROMPT]"
Is the African American English prompt a valid paraphrase of the Standard English prompt?

---

## A.5 IMPLEMENTATION DETAILS

### A.5.1 DATASET IMPLEMENTATION

For Algorithm, we unify the prompts by substituting all function names as python_function to avoid as much memorization as possible. We also manually corrected instances in HumanEval where the task descriptions were not precise enough (e.g., when the output data structure specified in the docstring is different from the one specified in the function heading). We also slightly modified some instructions in algorithm datasets without changing their intention to make sure our prompts are coherent (e.g., changing *to solve the following problem* to *to realize the following functionality*).

For other tasks, we unify the task output by asking LLMs to encode answers in $< answer ><$ $/answer >$ to enable easy parsing. All details can be found in ReDial dataset files.

### A.5.2 INFERENCE IMPLEMENTATION

We set temperature=0 and max new token as 4096 for all models at inference time unless specified in the main paper. We run experiments on GPT-4o/4/3.5 via Azure OpenAI service. We evaluate all other models via Azure Machine Learning Studio API for main results. Experiments run in the analysis part are hosted on 4 A100 with 80GB memory each.

## A.6 RESULTS FOR NON-ZERO TEMPERATURE

We vary the temperature by 0, 0.5, 0.7, and 1 on GPT-4o/4/3.5-turbo and Phi-3-Mini/Medium-128K-Instruct. When the temperature is not 0, we sample 3 answers per query and take average pass rates as results for corresponding settings. Results are in Figure 5.

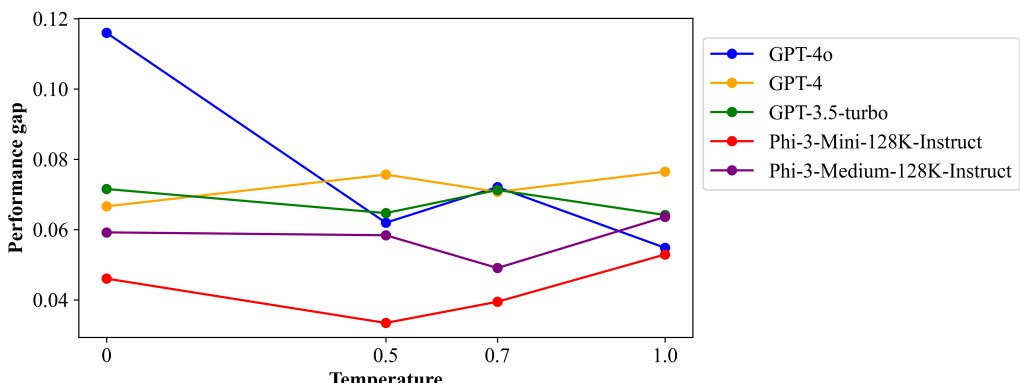

Figure 5: We vary the temperature by 0, 0.5, 0.7, 1 and report the performance gap between Standardized and AAVE ReDial.

We find that increasing temperature reduces the gap for GPT-4o in general, but does not affect other models' performance as much. Even when the performance gap is reduced, increasing temperature cannot cancel the gap.

## A.7 FULL RESULTS ON REDIAL

| Model | Setting | HumanEval | | MBPP | |
|---|---|---|---|---|---|
| | | Original | AAVE | Original | AAVE |
| GPT-4o 🔒 | Vanilla | 0.872 | $0.811_{(-)0.061}$ | 0.700 | $0.707_{(+)0.007}$ |
| | CoT | 0.841 | $0.805_{(-)0.037}$ | 0.693 | $0.713_{(+)0.02}$ |
| GPT-4 🔒 | Vanilla | 0.780 | $0.744_{(-)0.037}$ | 0.700 | $0.700_{(-)-0.0}$ |
| | CoT | 0.750 | $0.707_{(-)0.043}$ | 0.693 | $0.500_{(-)0.193}$ |
| GPT-3.5-turbo 🔒 | Vanilla | 0.640 | $0.622_{(-)0.018}$ | 0.667 | $0.640_{(-)0.027}$ |
| | CoT | 0.616 | $0.591_{(-)0.024}$ | 0.680 | $0.507_{(-)0.173}$ |
| LLaMA-3.1-70B-Instruct | Vanilla | 0.744 | $0.726_{(-)0.018}$ | 0.707 | $0.573_{(-)0.133}$ |
| | CoT | 0.738 | $0.689_{(-)0.049}$ | 0.707 | $0.613_{(-)0.093}$ |
| LLaMA-3-70B-Instruct | Vanilla | 0.689 | $0.671_{(-)0.018}$ | 0.673 | $0.613_{(-)0.06}$ |
| | CoT | 0.720 | $0.665_{(-)0.055}$ | 0.673 | $0.627_{(-)0.047}$ |
| LLaMA-3-8B-Instruct | Vanilla | 0.530 | $0.524_{(-)0.006}$ | 0.540 | $0.493_{(-)0.047}$ |
| | CoT | 0.537 | $0.512_{(-)0.024}$ | 0.527 | $0.440_{(-)0.087}$ |
| Mixtral-8x7B-Instruct-v0.1 | Vanilla | 0.402 | $0.390_{(-)0.012}$ | 0.507 | $0.413_{(-)0.093}$ |
| | CoT | 0.396 | $0.396_{(-)-0.0}$ | 0.547 | $0.427_{(-)0.12}$ |
| Mistral-7B-Instruct-v0.3 | Vanilla | 0.268 | $0.268_{(-)-0.0}$ | 0.400 | $0.240_{(-)0.16}$ |
| | CoT | 0.262 | $0.274_{(+)0.012}$ | 0.367 | $0.213_{(-)0.153}$ |
| Phi-3-Medium-128K-Instruct | Vanilla | 0.530 | $0.518_{(-)0.012}$ | 0.560 | $0.340_{(-)0.22}$ |
| | CoT | 0.530 | $0.573_{(+)0.043}$ | 0.567 | $0.327_{(-)0.24}$ |
| Phi-3-Small-128K-Instruct | Vanilla | 0.598 | $0.329_{(-)0.268}$ | 0.633 | $0.167_{(-)0.467}$ |
| | CoT | 0.585 | $0.293_{(-)0.293}$ | 0.553 | $0.087_{(-)0.467}$ |
| Phi-3-Mini-128K-Instruct | Vanilla | 0.549 | $0.482_{(-)0.067}$ | 0.567 | $0.367_{(-)0.2}$ |
| | CoT | 0.567 | $0.530_{(-)0.037}$ | 0.587 | $0.347_{(-)0.24}$ |

Table 5: All results for **Algorithm**.

| Model | Setting | Original | AAVE |
|---|---|---|---|
| GPT-4o 🔒 | Vanilla | 0.783 | $0.312_{(-)0.471}$ |
| | CoT | 0.762 | $0.662_{(-)0.1}$ |
| GPT-4 🔒 | Vanilla | 0.217 | $0.133_{(-)0.083}$ |
| | CoT | 0.283 | $0.058_{(-)0.225}$ |
| GPT-3.5-turbo 🔒 | Vanilla | 0.200 | $0.129_{(-)0.071}$ |
| | CoT | 0.075 | $0.067_{(-)0.008}$ |
| LLaMA-3.1-70B-Instruct | Vanilla | 0.392 | $0.113_{(-)0.279}$ |
| | CoT | 0.579 | $0.500_{(-)0.079}$ |
| LLaMA-3-70B-Instruct | Vanilla | 0.158 | $0.067_{(-)0.092}$ |
| | CoT | 0.517 | $0.350_{(-)0.167}$ |
| LLaMA-3-8B-Instruct | Vanilla | 0.025 | $0.067_{(+)0.042}$ |
| | CoT | 0.029 | $0.025_{(-)0.004}$ |
| Mixtral-8x7B-Instruct-v0.1 | Vanilla | 0.100 | $0.075_{(-)0.025}$ |
| | CoT | 0.133 | $0.071_{(-)0.062}$ |
| Mistral-7B-Instruct-v0.3 | Vanilla | 0.096 | $0.075_{(-)0.021}$ |
| | CoT | 0.083 | $0.083_{(-)-0.0}$ |
| Phi-3-Medium-128K-Instruct | Vanilla | 0.050 | $0.037_{(-)0.013}$ |
| | CoT | 0.067 | $0.029_{(-)0.037}$ |
| Phi-3-Small-128K-Instruct | Vanilla | 0.058 | $0.062_{(+)0.004}$ |
| | CoT | 0.096 | $0.079_{(-)0.017}$ |
| Phi-3-Mini-128K-Instruct | Vanilla | 0.021 | $0.042_{(+)0.021}$ |
| | CoT | 0.017 | $0.021_{(+)0.004}$ |

Table 6: All results for **Comprehensive**.

| Model | Setting | Folio | | LogicBench | |
|---|---|---|---|---|---|
| | | Original | AAVE | Original | AAVE |
| GPT-4o 🔒 | Vanilla | 0.938 | $0.870_{(-)0.068}$ | 0.720 | $0.685_{(-)0.035}$ |
| | CoT | 0.938 | $0.926_{(-)0.012}$ | 0.715 | $0.645_{(-)0.070}$ |
| GPT-4 🔒 | Vanilla | 0.858 | $0.796_{(-)0.062}$ | 0.745 | $0.710_{(-)0.035}$ |
| | CoT | 0.864 | $0.759_{(-)0.105}$ | 0.735 | $0.730_{(-)0.005}$ |
| GPT-3.5-turbo 🔒 | Vanilla | 0.605 | $0.519_{(-)0.086}$ | 0.475 | $0.565_{(+)0.090}$ |
| | CoT | 0.519 | $0.506_{(-)0.012}$ | 0.490 | $0.360_{(-)0.130}$ |
| LLaMA-3.1-70B-Instruct | Vanilla | 0.642 | $0.593_{(-)0.049}$ | 0.750 | $0.660_{(-)0.090}$ |
| | CoT | 0.870 | $0.827_{(-)0.043}$ | 0.760 | $0.720_{(-)0.040}$ |
| LLaMA-3-70B-Instruct | Vanilla | 0.673 | $0.623_{(-)0.049}$ | 0.655 | $0.495_{(-)0.160}$ |
| | CoT | 0.883 | $0.809_{(-)0.074}$ | 0.400 | $0.360_{(-)0.040}$ |
| LLaMA-3-8B-Instruct | Vanilla | 0.667 | $0.617_{(-)0.049}$ | 0.325 | $0.340_{(+)0.015}$ |
| | CoT | 0.599 | $0.660_{(+)0.062}$ | 0.375 | $0.355_{(-)0.020}$ |
| Mixtral-8x7B-Instruct-v0.1 | Vanilla | 0.327 | $0.401_{(+)0.074}$ | 0.485 | $0.110_{(-)0.375}$ |
| | CoT | 0.370 | $0.284_{(-)0.086}$ | 0.395 | $0.285_{(-)0.110}$ |
| Mistral-7B-Instruct-v0.3 | Vanilla | 0.481 | $0.537_{(+)0.056}$ | 0.180 | $0.055_{(-)0.125}$ |
| | CoT | 0.475 | $0.506_{(+)0.031}$ | 0.200 | $0.120_{(-)0.080}$ |
| Phi-3-Medium-128K-Instruct | Vanilla | 0.543 | $0.568_{(+)0.025}$ | 0.465 | $0.390_{(-)0.075}$ |
| | CoT | 0.698 | $0.574_{(-)0.123}$ | 0.325 | $0.330_{(+)0.005}$ |
| Phi-3-Small-128K-Instruct | Vanilla | 0.580 | $0.531_{(-)0.049}$ | 0.490 | $0.520_{(+)0.030}$ |
| | CoT | 0.728 | $0.568_{(-)0.160}$ | 0.395 | $0.485_{(+)0.090}$ |
| Phi-3-Mini-128K-Instruct | Vanilla | 0.420 | $0.352_{(-)0.068}$ | 0.755 | $0.665_{(-)0.090}$ |
| | CoT | 0.481 | $0.370_{(-)0.111}$ | 0.735 | $0.655_{(-)0.080}$ |

Table 7: All results for **Logic**.

| Model | Setting | GSM8K | | SVAMP | |
|---|---|---|---|---|---|
| | | Original | AAVE | Original | AAVE |
| GPT-4o 🔒 | Vanilla | 0.933 | $0.947_{(+)0.013}$ | 0.933 | $0.913_{(-)0.020}$ |
| | CoT | 0.967 | $0.933_{(-)0.033}$ | 0.933 | $0.907_{(-)0.027}$ |
| GPT-4 🔒 | Vanilla | 0.840 | $0.640_{(-)0.200}$ | 0.840 | $0.787_{(-)0.053}$ |
| | CoT | 0.947 | $0.867_{(-)0.080}$ | 0.893 | $0.760_{(-)0.133}$ |
| GPT-3.5-turbo 🔒 | Vanilla | 0.587 | $0.287_{(-)0.300}$ | 0.747 | $0.600_{(-)0.147}$ |
| | CoT | 0.780 | $0.480_{(-)0.300}$ | 0.727 | $0.607_{(-)0.120}$ |
| LLaMA-3.1-70B-Instruct | Vanilla | 0.680 | $0.920_{(+)0.240}$ | 0.853 | $0.867_{(+)0.013}$ |
| | CoT | 0.867 | $0.927_{(+)0.060}$ | 0.893 | $0.813_{(-)0.080}$ |
| LLaMA-3-70B-Instruct | Vanilla | 0.933 | $0.920_{(-)0.013}$ | 0.880 | $0.853_{(-)0.027}$ |
| | CoT | 0.947 | $0.907_{(-)0.040}$ | 0.900 | $0.867_{(-)0.033}$ |
| LLaMA-3-8B-Instruct | Vanilla | 0.847 | $0.800_{(-)0.047}$ | 0.807 | $0.800_{(-)0.007}$ |
| | CoT | 0.820 | $0.800_{(-)0.020}$ | 0.833 | $0.800_{(-)0.033}$ |
| Mixtral-8x7B-Instruct-v0.1 | Vanilla | 0.427 | $0.193_{(-)0.233}$ | 0.613 | $0.487_{(-)0.127}$ |
| | CoT | 0.673 | $0.573_{(-)0.100}$ | 0.700 | $0.560_{(-)0.140}$ |
| Mistral-7B-Instruct-v0.3 | Vanilla | 0.367 | $0.147_{(-)0.220}$ | 0.433 | $0.280_{(-)0.153}$ |
| | CoT | 0.420 | $0.320_{(-)0.100}$ | 0.487 | $0.373_{(-)0.113}$ |
| Phi-3-Medium-128K-Instruct | Vanilla | 0.893 | $0.833_{(-)0.060}$ | 0.840 | $0.747_{(-)0.093}$ |
| | CoT | 0.893 | $0.853_{(-)0.040}$ | 0.827 | $0.800_{(-)0.027}$ |
| Phi-3-Small-128K-Instruct | Vanilla | 0.840 | $0.793_{(-)0.047}$ | 0.800 | $0.727_{(-)0.073}$ |
| | CoT | 0.880 | $0.873_{(-)0.007}$ | 0.907 | $0.813_{(-)0.093}$ |
| Phi-3-Mini-128K-Instruct | Vanilla | 0.520 | $0.573_{(+)0.053}$ | 0.520 | $0.527_{(+)0.007}$ |
| | CoT | 0.800 | $0.807_{(+)0.007}$ | 0.747 | $0.693_{(-)0.053}$ |

Table 8: All results for **Math**.

## A.8  MULTIVALUE PERTURBATION

Since the unfamiliarity of data cannot explain the whole picture, how much can we attribute the failure to AAVE-specific features? We use the rule-based transformation method in Ziems et al. (2023) to inject AAVE features into our dataset for synthetic probing. We compare GPT-4o/4/3.5 and Phi-3-Medium/Mini-128k-Instruct performance in feature densities of $\{0, 0.25, 0.5, 0.75, 1\}$ and run the same setting as the main experiment.

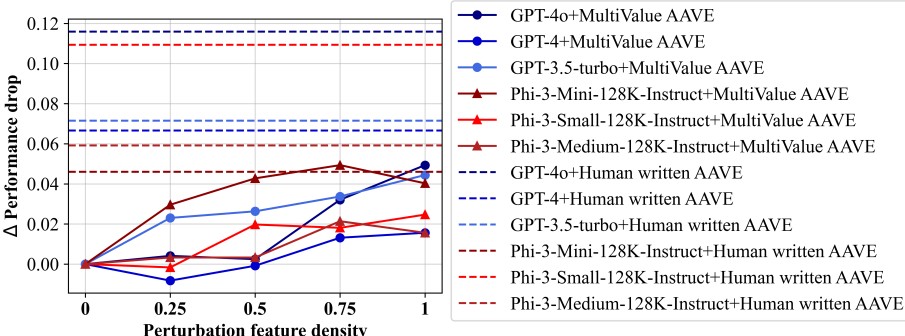

Figure 6: Perturbation with AAVE features. We control perturbation feature densities at $\{0, 0.25, 0.5, 0.75, 1\}$ to gradually inject AAVE features using rule-based transformations.

Results are shown in Figure 6. On the one hand, we find that models generally show increasing performance drops with increasing feature density, which means that AAVE-specific features do contribute to model performance drops. On the other hand, even drops caused by the strongest perturbation are generally far from the drops caused by human-rewritten prompts. This shows the limitation of previous methods in revealing LLM robustness based on synthetic data as there can be more influential factors than what lexico-syntactic rules can capture. Phi-3-Mini-128K-Instruct is again an outlier here, being that it is the only model that has a stronger performance drop in feature injections compared to human-written dialect data.

