# OpenReview forum: "One Language, Many Gaps: Evaluating Dialect Fairness and Robustness of Large Language Models in Reasoning Tasks"
_ICLR.cc/2025/Conference — ICLR 2025 Conference Withdrawn Submission_

### Official Review · Reviewer_Y4E1 · 2024-10-24

**Soundness:** 2
**Presentation:** 3
**Contribution:** 2
**Rating:** 3
**Confidence:** 4

**Summary:**

This work proposed a new benchmark: ReDial, the human-annotated benchmark dataset that evaluates the fairness and robustness of large language models (LLMs) when handling African American Vernacular English (AAVE) dialect. This benchmark contains questions including algorithmic, mathematical, logical, and comprehensive reasoning, and compares model performance on parallel Standardized English and AAVE prompts.

The study finds that most LLMs and most types of questions, show significant brittleness when dealing with AAVE, performing substantially worse than with Standardized English. Additionally, simply asking models to rephrase queries in Standardized English fails to close the performance gap, often increasing computational costs. The authors argue that this dialectal unfairness reflects deeper issues in how LLMs are trained and evaluated.

**Strengths:**

1. This work provides the first comprehensive evaluation of large language models (LLMs) on dialect fairness and robustness, specifically focusing on African American Vernacular English (AAVE), which is a contribution to NLP fairness and LLM literature.

2. This paper introduces the ReDial dataset, a benchmark in AAVE and Standardized English, which covers different domains such as algorithm, logic, math, and comprehensive reasoning tasks. This new dataset is valuable for future research on dialect robustness in reasoning tasks.

3. The paper conducts extensive experiments on multiple current LLMs, such as GPT-4, LLaMA, and Mistral, showing their brittleness when handling AAVE, thereby providing strong empirical evidence of dialectal bias in complex reasoning tasks.

**Weaknesses:**

1. The poor performance of models on non-standard English, especially AAVE, is somewhat expected, given the dialectal imbalance in training data. While the paper provides empirical evidence, the findings don't go beyond what could be anticipated, limiting its novelty.

2. This paper points out the weakness of LLM when handling AAVE but doesn’t propose any effective solutions to mitigate these issues. However, it mentions that simple data augmentation isn’t enough. Providing alternative strategies such as new architectures or advanced data augmentation techniques would make this work more strong.

3. Since the main contribution of this paper is the creation of the ReDial dataset and its evaluation, a benchmark track rather than the main conference would be a suitable place.

4. Although the paper covers algorithms, logic, math, and comprehensive reasoning tasks. There are still standardized methods to evaluate fairness. I feel for fairness evaluation, those methods are worth trying.

5. The paper demonstrates the performance degradation on AAVE reasoning tasks, but it does not dive deeply into which specific linguistic features of AAVE (e.g., vocabulary, syntax) cause model confusion. This leaves a gap in the theoretical understanding of the problem. Providing more insight would be better.

**Questions:**

1. Could further experiments distinguish the impact of model architecture and training strategies on the performance drop? For example, LLaMA 3-8B shows a smaller performance drop on AAVE, while similar-sized models like Phi-3 Small experience a significant drop. Does this suggest that different model architectures handle language variations differently?

2. Model size doesn’t seem to be the sole factor—larger models don’t always mitigate the performance drop on AAVE. For example, Mistral and Phi-3 Small models still experience significant performance degradation. Does this imply that task-specific model optimization is more crucial?

3. GPT-4o’s errors in rephrasing may indicate a lack of fine-grained understanding of AAVE semantics and syntax. Does this suggest that the model’s knowledge is insufficient for AAVE, further explaining its weaker performance on AAVE tasks？

4. Could further fine-tuning or training with a dataset containing more dialectal data improve models’ performance on AAVE? Specifically, could models that struggle with rephrasing tasks benefit from additional exposure to dialects?

5. Is 1.2K parallel sentence pairs enough to cover the diversity of AAVE and represent the real-world usage of the AAVE community? Would a more diverse and larger dataset be necessary to further validate the model's performance?

---

> ### Author Response · Authors · 2024-11-18
>
> **We would deeply appreciate it if you could read the general comment we wrote at the top of the rebuttal page.**
>
> Thank you for acknowledging the novelty of our task, the comprehensive design of our dedicatedly curated dataset, and the extensive experiments on a wide range of LLMs. We will address the weaknesses and questions you raised below.

---

> ### Author Response · Authors · 2024-11-18
> **Regarding the weaknesses:**
>
> *W1: …findings don't go beyond what could be anticipated, limiting its novelty…*
>
> We do not believe LLMs’ unfair treatment to any specific demographic groups should be anticipated. Even if people assume underperformance to minorities, the problem should not be neglected or considered not novel. **The novelty of our study lies in pointing out an important yet long-ignored problem and conducting a comprehensive experiment to reveal this disappointing fact. Moreover, no existing work has shown this in reasoning.** The fact that the model can't do certain tasks in a dialect doesn't mean it struggles for all tasks.
>
>
> *W2: Providing alternative strategies such as new architectures or advanced data augmentation techniques would make this work more strong…*
>
> We agree with the reviewer that it would be ideal to immediately provide a solution to the problem. In fact, **we tried standardized prompting to reduce the gap, but we found that it is not easily cancellable. We think this is especially interesting and provides a direction for many future studies.**
>
>
> *W3: …a benchmark track rather than the main conference would be a suitable place*
>
> To the best of our knowledge, there is no specific benchmark track in ICLR. In addition, **we do see similar benchmark papers published in ICLR main conference previously; for example, MMLU**, one of the most important benchmarks in use today, was published at ICLR 2021 [1].
>
> *W4: Although the paper covers algorithms, logic, math, and comprehensive reasoning tasks. There are still standardized methods to evaluate fairness. I feel for fairness evaluation, those methods are worth trying.*
>
> We note that LLMs are well-known to be more or less biased in previous benchmarks such as StereoSet [2], CrowS-Pairs [3], and BBQ [4]. **In ReDial, we find a completely new manifestation of the bias.**
>
> *W5: …which specific linguistic features of AAVE (e.g., vocabulary, syntax) cause model confusion...*
>
> According to our synthetic probing experiment in Appendix A.8, **morphosyntactic AAVE features indeed impact LLM performance**. However, we find that LLMs generally show a much stronger performance drop on real human-annotated data compared to synthetically generated data, which indicates there are potentially other impacting factors.
>
> [1] Hendrycks, D., Burns, C., Basart, S., Zou, A., Mazeika, M., Song, D., & Steinhardt, J. Measuring Massive Multitask Language Understanding. In International Conference on Learning Representations.
>
> [2] Nadeem, M., Bethke, A., & Reddy, S. (2020). StereoSet: Measuring stereotypical bias in pretrained language models. arXiv preprint arXiv:2004.09456.
>
> [3] Nangia, N., Vania, C., Bhalerao, R., & Bowman, S. R. (2020). CrowS-pairs: A challenge dataset for measuring social biases in masked language models. arXiv preprint arXiv:2010.00133.
>
> [4] Parrish, A., Chen, A., Nangia, N., Padmakumar, V., Phang, J., Thompson, J., ... & Bowman, S. R. (2021). BBQ: A hand-built bias benchmark for question answering. arXiv preprint arXiv:2110.08193.

---

> ### Author Response · Authors · 2024-11-18
> **Regarding your questions:**
>
> *Q1: Could further experiments distinguish the impact of model architecture and training strategies on the performance drop? For example, LLaMA 3-8B shows a smaller performance drop on AAVE, while similar-sized models like Phi-3 Small experience a significant drop…*
>
> **We suspect this is mainly caused by training data differences.** Both models are dense transformers but Phi models use highly curated synthetic data (i.e., textbook data) in their pre-training, which probably results in their brittleness when handling non-standard dialect inputs.
>
> *Q2: Model size doesn’t seem to be the sole factor… Does this imply that task-specific model optimization is more crucial?*
>
> We do not consider models should be tuned for specific dialect tasks. After all, **there is no highly accurate way to detect user dialect to LLMs, which makes it difficult to route to task-specific models even if we have them**.
>
> *Q3: …Does this suggest that the model’s knowledge is insufficient for AAVE, further explaining its weaker performance on AAVE tasks?*
>
> **We refuted this hypothesis in section 4.1**, by perturbing standardized English data until LLMs find it even more confusing than AAVE. We find that LLMs show worse performance in AAVE even when they understand AAVE better than typo-ed standardized English.
>
> *Q4: …fine-tuning or training with a dataset containing more dialectal data improve models’ performance on AAVE…?*
>
> There are three main reasons for why we didn’t attempt to fine-tune models. First, there is no immediately available dialect dataset for reasoning fine-tuning. Second, even if fine-tuning is effective, **we do not expect most end users to fine-tune their models instead of simply interacting with chatbots**. Last, the main contribution of our work is to point out an important yet long-ignored problem, and we want to leave this question for further work.
>
> *Q5: …Is 1.2K parallel sentence pairs enough…? Would a more diverse and larger dataset be necessary…?*
>
> **ReDial is a benchmark aimed at testing instead of fine-tuning. In fact, a benchmark of 1.2k datapoints is already pretty big (consider that HumanEval has 164 datapoints, the unsanitized MBPP has 1k, and GSM8K has 1.3k), especially considering the fact that it is end-to-end human-annotated.** The expense of recruiting expert dialect annotators is enormous, and in our opinion, there is no need to collect more data.

---

> ### Comment · Reviewer_Y4E1 · 2024-11-28
> **Reply to authors**
>
> Thank you for your reply, I will keep my original score. I still think this contribution may not be suitable for the ICLR main track. The Benchmark Track or *CL might be a more appropriate choice.

---

### Official Review · Reviewer_CTv9 · 2024-10-30

**Soundness:** 2
**Presentation:** 3
**Contribution:** 2
**Rating:** 5
**Confidence:** 3

**Summary:**

African American Vernacular English (AAVE) is a dialect less frequently seen in the training dataset.
This paper proposed to assess the bias and robustness of LLMs thru a proposed benchmark, namely ReDial.
With native AAVE speakers' validation, ReDial covers 4 category reasoning tasks, including algo, math, logic, and comprehension.
The benchmark is then tested against GPT, LLaMA, Mistral, and Phi-3 models.
AAVE queries showed a suboptimal performance compared to standard English queries in various aspects.

**Strengths:**

- The paper is nicely organized and easy to follow.
- Code is publicly available.
- The construction of the benchmark Redial is well presented, including the source benchmark of sampling and native speaker validation.
  - To ensure the quality of algo translation, annotators have a CS background.
  - The validation process employed an iterative approach for better quality and limited the usage of LLMs during the process
- Experiment covered a wide of range of popular models and investigate two prompting methods (zero-shot & zero-shot CoT)
- Appendix is informative

**Weaknesses:**

- Using dialects to reduce bias and improve robustness of LLMs sounds like an exciting direction. ReDial (Reasoing with Dialect Queries) currently contains one dialect, AAVE, which may appear to be limited. Expanding the scope to include a diverse of dialects may more comprehensively capture the insight of dialects in general when assessing LLMs bias and robustness.

- AAVE is widely spoken and has a large amount of data sources. What about other dialects sharing similar characteristics?

- Non-standard English dialect is semantically equivalent to the original English questions, offering similar insights to previous study on the impact of multilingual and tone on LLMs performance.

- Table 3 showed the detailed comparison in performance among the four categories. However, it is not clear about which model is used. And other models' results were not included in the appendix.

- The submission has one extra page. Refers to "There will be a strict upper limit of 9 pages for the main text of the submission, with unlimited additional pages for citations. "
- The claim that the daily use of LLMs is limited to zero shot, and zero shot chain-of-thought is unsound. (line 252)

- Perplexity is used as an indication of LLM's understanding of data. After injection of typos into standardized English ReDial, the perplexity of it exceeded the AAVE one. This may not reveal the fairness directly because the increased perplexity may reflect model's sensitivity to surface-level variations, in this case, typos, instead of a deeper understanding towards dialects.

- Compared to increasing perplexity of standardized English ReDial, finetuning process could be adopted to improve the knowledge of a dialect and consequently reduce the perplexity of AAVE.

**Questions:**

- Have you attempted to fine-tune any models using the ReDial dataset? Given that fine-tuning could enhance the model’s understanding of the dataset, is it expected to improve the performance of LLMs?

- Have you tried to use few-shot prompt for in-context learning and thus potentially better performance?

- Did you look into why GPT-4o & 3.5 turbo see a performance degradation with Chain-of-Thought prompting in original English questions? It may not align with prior research findings.

---

> ### Author Response · Authors · 2024-11-18
>
> **We would deeply appreciate it if you could read the general comment we wrote at the top of the rebuttal page.**
>
> Thank you for acknowledging the clear and informative presentation of the paper, the availability of code, the comprehensive evaluation of models, and the significance of our carefully curated dataset. We will address the weaknesses and questions you raised below.

---

> ### Author Response · Authors · 2024-11-18
> **Regarding weaknesses:**
>
> *W1: …ReDial (Reasoing with Dialect Queries) currently contains one dialect, AAVE, which may appear to be limited…*
>
> We note that non-standardized dialect data collection, compared to standardized dialect, is a non-trivial task. The population that speaks non-standardized dialect tends to be limited, and it is especially difficult to find among them the domain experts who are qualified to rewrite our reasoning benchmark. Meanwhile, we consider **our pioneering ReDial to set up a standard pipeline for the future development of relevant datasets**: it makes the work a lot easier as future work can just draw on our pipeline, so extension to other dialects should be straightforward. We hope the extension to more dialects to be the result of a community effort.
>
>
> *W2: AAVE is widely spoken and has a large amount of data sources. What about other dialects sharing similar characteristics?*
>
> We agree with the reviewer that it would be interesting if we could draw a general conclusion. Unfortunately, it is not immediately doable. **Our work is pioneering in the sense that no existing English dialect work provides the same natural data in reasoning tasks**, to the best of our knowledge. Therefore, we refrain from making claims for other dialects.
>
>
> *W3: (this study)...offering similar insights to previous study on the impact of multilingual and tone on LLMs performance…*
>
> As far as we know, there do not seem to be extremely similar works to ours in multilingual and tone studies that offer the same insight.
>
> *W4: Table 3 … is not clear about which model is used. And other models' results were not included in the appendix.*
>
> Table 3 results are averaged across models. Further details are in Appendix A.7, where we present all fine-grained results.
>
> *W5: The submission has one extra page. Refers to "There will be a strict upper limit of 9…*
>
> ICLR allows 10 pages this year. Refers to “New this year, the main text must be between 6 and 10 pages (inclusive). This limit will be strictly enforced. Papers with main text on the 11th page will be desk rejected. The page limit applies to both the initial and final camera ready version.”
>
> *W6: The claim that the daily use of LLMs is limited to zero shot, and zero shot chain-of-thought is unsound…*
>
> Indeed, we think it is most likely that most users, and especially laypersons only use zero shot in general cases. **As in general using scenario, it is less practical for end-users to find highly releated and fine-designed few-shot examples.** Meanwhile, not all LLM users are experts in prompting LLM. So we only included CoT in the expectation that expert users will know about this technique, and that many models have built-in CoT reasoning now.
>
> *W7: Perplexity is used as an indication of LLM's understanding of data…This may not reveal the fairness directly because the increased perplexity may reflect model's sensitivity to surface-level variations...*
>
> Perplexity has long been used in important works to measure language familiarity. Please refer to [1] inter alia.
>
> *W8: …finetuning process could be adopted…*
>
> We note that the key of this paper is not in finding a solution with complicated tuning process per se: for instance, even if fine-tuning is effective, **we do not expect most of end users to fine-tune their models instead of simply interacting with chatbots**. Therefore, the fact that these models are biased is in and of itself relevant.
>
> [1] Gonen, H., Iyer, S., Blevins, T., Smith, N. A., & Zettlemoyer, L. (2022). Demystifying prompts in language models via perplexity estimation. arXiv preprint arXiv:2212.04037.

---

> > ### Comment · Reviewer_CTv9 · 2024-11-28
> >
> > A standardized pipeline is a solid foundation; however, if this is presented as the main contribution of the paper, it may fall short in terms of novelty.
> >
> > > "we think it is most likely that most users, and especially laypersons only use zero shot in general cases"
> >
> > It would be beneficial to explore and experiment with different prompting methods in your research to enhance correctness and identify potential improvements. Additionally, the claim that users exclusively use direct prompting should be supported with evidence or user studies.
> >
> > > "we do not expect most of end users to fine-tune their models instead of simply interacting with chatbots"
> >
> > If the paper does not aim to solve the fairness issue, framing it as a dataset track could provide more clarity and align better with the paper's focus.

---

> ### Author Response · Authors · 2024-11-18
> **Regarding the questions:**
>
> *Q1: …fine-tune any models using the ReDial dataset…*
>
> ReDial is only for testing but not tuning. **We do not want to tune the model on ReDial as it would incur severe problems of “train on test”.** For why we didn’t finetune with other datasets, please refer to answers to W8 above.
>
> *Q2: …few-shot prompt for in-context learning…*
>
> Our work aims at mimicking general use cases, and **we do not strongly believe that we should ask/expect users to use complicated prompt engineering skills when they query chatbots**.
>
> *Q3: …performance degradation with Chain-of-Thought prompting in original English questions? It may not align with prior research findings…*
>
> First, **there are some works showing on CoT not helping and even degrading LLM performance (see [2] inter alia), and our observation aligns with their findings.** Second, the CoT degradation is very minimal, **not even statistically significant** in our experiments, which is likely to be caused by some random effects. We will add a discussion to the final version of the paper.
>
> [2] Liu, R., Geng, J., Wu, A. J., Sucholutsky, I., Lombrozo, T., & Griffiths, T. L. (2024). Mind your step (by step): Chain-of-thought can reduce performance on tasks where thinking makes humans worse. arXiv preprint arXiv:2410.21333.

---

### Official Review · Reviewer_S61L · 2024-10-31

**Soundness:** 2
**Presentation:** 3
**Contribution:** 2
**Rating:** 3
**Confidence:** 3

**Summary:**

The paper presents a novel study on the fairness and robustness of LLMs when dealing with dialects, specifically African American Vernacular English (AAVE). The authors have created a benchmark dataset, ReDial, comprising over 1.2K parallel query pairs in Standardized English and AAVE to evaluate LLMs on reasoning tasks. The study finds that most LLMs show significant performance degradation on AAVE queries compared to Standardized English, indicating a lack of fairness and robustness.

**Strengths:**

The paper addresses a critical and underexplored issue in the field of natural language processing, namely the fairness and robustness of LLMs to dialectal variations within a language.

The creation of ReDial, a high-quality, human-annotated dataset for evaluating LLMs on reasoning tasks in AAVE, is a significant contribution to the research community.

The paper is well-organized, and the arguments are presented clearly, making it easy to follow the authors' reasoning and conclusions.

**Weaknesses:**

The paper attempts to explore potential reasons behind the observed performance degradation with AAVE, but it merely dismisses data skewness without offering alternative explanations.

In lines 343-372, 4 observations emerge from the experiments.
(1) All models exhibit fragility when handling AAVE.
(2) All reasoning tasks demonstrate vulnerability to AAVE.
(3) Increasing model size does not enhance robustness against AAVE.
(4) Highly curated datasets are particularly susceptible to AAVE.
Observations (1) to (3) are acceptable. However, observation (4) contradicts the findings in Table 2, where LLaMA-3-8B-Instruct shows the least performance decline. In comparison, Mistral and GPT exhibit greater drops than LLaMA-3-8B-Instruct, making observation (4) questionable.

Is data skewness truly irrelevant to AAVE fragility? The experiment lacks persuasiveness. As illustrated in lines 415-419, the experiment simulates typographical errors by altering characters in Standardized ReDial and compares its performance to AAVE. Introducing typographical errors in Standardized ReDial increases perplexity. However, higher perplexity signifies greater uncertainty in token prediction. It does not accurately measure language familiarity, as illogical sentence structures can also result in high perplexity. Thus, it is inappropriate to compare this with AAVE.

**Questions:**

What are the theoretical underpinnings of the performance gap between Standardized English and AAVE in LLMs?

How do the findings generalize to other dialects or non-English languages?

Can the authors propose any solutions or strategies to mitigate the performance gap and improve fairness for dialect speakers?

---

> ### Author Response · Authors · 2024-11-18
>
> **We would deeply appreciate it if you could read the general comment we wrote at the top of the rebuttal page.**
>
> Thank you for acknowledging the importance and novelty of our study, the significant contribution of our dataset to the community, and the clear presentation of the paper. We will address the weaknesses and questions you raised below.

---

> ### Author Response · Authors · 2024-11-18
> **Regarding Weaknesses:**
>
> *W1: …it merely dismisses data skewness without offering alternative explanations…*
>
> We would like to emphasize that **this is the most interesting finding** in the paper: **people tend to believe that the lack of training data is the one and only reason that’s causing problems** in non-standard data distribution. However, **we dispute this hypothesis** by running comprehensive experiments to prove that this is not the major factor. To the best of our knowledge, there are no immediately available alternative statistical explanations for this problem. This observation creates many interesting potential future questions for the community to explore.
>
> In fact, we think it is especially interesting that we do not have an immediate clear explanation here. We tried to explain it with perplexity (i.e., language familiarity/data skewness) but it does not seem to be explaining what’s going on. We think it will create many interesting directions for future works to follow.
>
>
> *W2: …(4) Highly curated datasets are particularly susceptible to AAVE… Mistral and GPT exhibit greater drops than LLaMA-3-8B-Instruct, making observation (4) questionable…*
>
> **Mistral/GPT being more brittle than LLaMA-3 does not invalidate our claim.** To the best of our knowledge, **only Phi models explicitly state that they use synthetic data in pre-training** [1]. LLaMA-3 only uses synthetic data in post-training [2], so the model being more robust than GPT/Mistral does not seem to give much evidence to the effectiveness of using highly curated data in pre-training. We will be more cautious about the statement and revise it in the later version. Thanks for pointing it out.
>
> *W3: …higher perplexity signifies greater uncertainty in token prediction. It does not accurately measure language familiarity, as illogical sentence structures can also result in high perplexity…*
>
> We agree that perplexity can be interpreted as uncertainty, but there is more than one interpretation of perplexity. Language models are trained with the objective of minimizing entropy in token predictions. With lower entropy, the uncertainty is of course lower. At the same time, the reason why language models can have low uncertainty is because the language model gets increasingly familiar with certain language expressions during training. In fact, it is not surprising that illogical sentence structures have high perplexity, presumably because humans use them less often than logical structures, which results in language models being less familiar with illogical constructions. Furthermore, **the method of using perplexity to estimate language familiarity that we follow is a long-established, tested practice in NLP literature (see [3] inter alia)**.
>
> [1] Gunasekar, S., Zhang, Y., Aneja, J., Mendes, C. C. T., Del Giorno, A., Gopi, S., ... & Li, Y. (2023). Textbooks are all you need. arXiv preprint arXiv:2306.11644.
>
> [2] Dubey, A., Jauhri, A., Pandey, A., Kadian, A., Al-Dahle, A., Letman, A., ... & Ganapathy, R. (2024). The llama 3 herd of models. arXiv preprint arXiv:2407.21783.
>
> [3] Gonen, H., Iyer, S., Blevins, T., Smith, N. A., & Zettlemoyer, L. (2022). Demystifying prompts in language models via perplexity estimation. arXiv preprint arXiv:2212.04037.

---

> > ### Comment · Reviewer_S61L · 2024-11-26
> >
> > Thank you for the feedback. While the clarifications are provided, the case remains unconvincing.
> >
> > It is not surprising that Standardized ReDial with injected typos outperforms AAVE. Despite the presence of typos, LLMs can still understand the input because they are trained on vast datasets where typos are common. Conversely, AAVE presents a more significant challenge due to its underrepresentation in training datasets, which increases the difficulty for LLMs to accurately interpret it.
> >
> > From this point, we can not get the conclusion directly: naively increasing AAVE in the training data may not diminish the performance gap.
> >
> > Regarding the contribution of this paper, you find a problem, and this problem is not surprising. What would be valuable is an exploration of why this phenomenon occurs and potential solutions to address it. Unfortunately, the paper does not delve into these areas.

---

> ### Author Response · Authors · 2024-11-18
> **Regarding your questions:**
>
> *Q1: What are the theoretical underpinnings of the performance gap between Standardized English and AAVE in LLMs?*
>
> Please refer to W1.
>
>
> *Q2: How do the findings generalize to other dialects or non-English languages?*
>
> We do not know if the finding can be transferred in the exact way to other dialects/non-English languages. After all, we do not know what is the reason that is causing the gap. It will be interesting to raise unified arguments for these dialects/non-English languages, but we suspect it will be very difficult, if not impossible, considering all the differences in these variations (e.g. [4, 5], inter alia).
>
>
> *Q3: Can the authors propose any solutions or strategies to mitigate the performance gap and improve fairness for dialect speakers?*
>
> There is no easy solution as we observe. We tried prompt-based debiasing method but it didn’t cancel the gap while introduced more cost. **Most importantly, we think proposing a solution is not the main point of our work: we spent considerable amount of efforts to curate a high-quality dataset, which can serve as a testbed to answer many relevant interesting questions and improve the technology in the future.**
>
>
> [4] Huang, H., Tang, T., Zhang, D., Zhao, W. X., Song, T., Xia, Y., & Wei, F. (2023). Not all languages are created equal in llms: Improving multilingual capability by cross-lingual-thought prompting. arXiv preprint arXiv:2305.07004.
>
> [5] Petrov, A., La Malfa, E., Torr, P., & Bibi, A. (2024). Language model tokenizers introduce unfairness between languages. Advances in Neural Information Processing Systems, 36.

---

### Official Review · Reviewer_XKeN · 2024-11-04

**Soundness:** 3
**Presentation:** 3
**Contribution:** 2
**Rating:** 6
**Confidence:** 4

**Summary:**

I had wrongly submitted a review for another paper - I have now corrected this - I apologise!

This paper explores how robust language models are to language variation and dialects. They create a test set with 1.2k query pairs in Standard English and African American Vernacular English (AAVE). They show that a variety of standard large language models perform worse on AAVE than on standard English, and on misspelled English. Prompting the LLM to rephrase quieries in Standard English improves performance but does not close the gap in performance. This is a nice contribution to the field and the dataset could be of general interest.

**Strengths:**

Nice dataset for measuring LLMs robustness to dialects Study showing brittleness of models to AAVE

**Weaknesses:**

Lack of discussion about the appropriateness of AAVE in different communications scenarios/tasks I would have expected some more experiments investigating different robustness techniques for LLMs to mitigate the problem with dialects

**Questions:**

Have the authors considered that people switch registers depending on pragmatics, and that a AAVE speaker when interacting via text with an LLM might naturally switch to something closer to Standard English by default?

Also some of the datasets used do not neatly fit into the type of problems that would require a lot of variation in vernacular eg. maths problems would be quite similar no matter what vernacular used and using a very different query for a maths problem in AAVE might be artificial and forced. The examples in Figure 1 do seem to be forced (especially the first three) and I think an AAVE speaker would feel like they are forcing a more significant difference than would be natural. I realise you have a naturalness check in the annotation pipeline, but this could just be that people thought it would plausibly be AAVE not that it should be AAVE - if annotators are given instruction to rewrite in AAVE they might do so even it is not very authentic. Some discussion of AAVE being different under different communication scenarios and tasks would be welcomed.

It was not clear if the answers were also rewritten in AAVE or just the queries as the description says it is just the queries, but the answers in AAVE would also be useful, measuring generation capability as well as understanding and reasoning.

---

> ### Author Response · Authors · 2024-11-13
>
> Dear Reviewer XKeN,
>
> Thank you for your review. It seems that your comments may not pertain to our paper. Could you kindly update your review to reflect our submission?  We will promptly address your concerns and look forward to further discussion with you.
>
> Thank you for your time and attention!

---

> ### Author Response · Authors · 2024-11-20
> **Regarding Weaknesses**
>
> Thank you for acknowledging the importance of our dataset and significance of our study! We will address the weaknesses and questions you have below.
>
> *W1: Lack of discussion about the appropriateness of AAVE in different communications scenarios/tasks…*
>
> We agree that we should discuss further about the different styles of AAVE in different scenarios/tasks in the limitation part. We will add relevant discussions in later version of the paper.
>
> *W2: …more experiments investigating different robustness techniques for LLMs to mitigate the problem with dialects*
>
> We agree that it would be interesting to explore more techniques. We tried prompting LLMs to rephrase in Standardized English first, but we found that this method does not cancel the gap while tends to introduce more cost. Besides, we consider this paper to be the first step towards evaluating LLMs’ dialect robustness in reasoning tasks. We hope future works can explore more methods to improve LLM robustness in dialects.

---

> ### Author Response · Authors · 2024-11-20
> **Regarding your questions**
>
> *Q1: Have the authors considered that people switch registers depending on pragmatics…*
>
> We agree that we should include more discussions about people switching styles in different pragmatic scenarios. However, we note that **the instructions we gave to our annotators were telling them to rewrite in a way that is natural to the dialect’s speakers instead of trying to collect as many distinct dialect features as possible**. Besides, although there is no clear evidence that people are using exactly the same register in prompting chatbots as we describe, there is no clear evidence that people do not use this register either. Therefore, we consider the annotation to be reflective of the way people actually use these queries.
>
> *Q2: some of the datasets used do not neatly fit into the type of problems that would require a lot of variation in vernacular eg. maths problems…this could just be that people thought it would plausibly be AAVE not that it should be AAVE …*
>
> We note that **all of the tasks we are considering are expressed in natural language**, which offers us a good testbed to simulate real use scenarios with dialect features. Further, we note that our dataset is fully end to end annotated by humans first, then checked for naturalness. We believe that the first annotation stage is already ensuring some degree of naturalness, while most of other related works as discussed in our paper generate dialect data by algorithms/LLMs and only have human in the loop in quality check.
>
> *Q3: not clear if the answers were also rewritten in AAVE… but the answers in AAVE would also be useful…*
>
> We agree that it would be ideal if we could also evaluate dialect generation quality. However, we expect this to be the next step as automated evaluation of dialect generation quality requires much more effort, which is beyond the scope of our paper. Our answers are not in AAVE as we only evaluate numerical values which offers us an objective measurement of the correctness of the answers instead of relying on LLMs as evaluators which can be expensive and sometimes biased. In addition, there are some relevant works in the manual evaluation of non-standard dialect generation quality, which makes us consider that there is no immediate need to do further evaluations in this aspect [1].
>
> [1] Fleisig, E., Smith, G., Bossi, M., Rustagi, I., Yin, X., & Klein, D. (2024). Linguistic Bias in ChatGPT: Language Models Reinforce Dialect Discrimination. arXiv preprint arXiv:2406.08818.

---

### Author Response · Authors · 2024-11-18
**General reply**

Dear reviewers,

Thank you for acknowledging the **novelty and importance** of our study, the significant contribution as well as the future impact of **our dataset ReDial** to the community as it **serves as the first step towards objectively measuring linguistic inclusivity for non-standard dialect users in reasoning tasks, especially considering there are lots of fairness efforts on multilingual but rarely in non-standard dialect**.

**The main weaknesses concerning the reviewers, i.e., that we cannot easily explain or mitigate the unfairness gap, are exactly the reasons why we consider this study of particular value to the research community. If we cannot easily explain or solve a problem, it means that the problem is worth special attention.** In fact, we have tried to develop engineering methods and explain the performance gap as a lack of familiarity/understanding on the side of the LLM, but we find that both of these approaches fall short. We would like to emphasize that the main contribution of this paper is to raise an important question and provide an easy-to-use framework for future studies. We understand that the reviewers think it would be ideal if we could solve problems when we suggest them, but we note many important scientific advancements start with raising questions (e.g., [1, 2]).

For the fairness study, we want to emphasize that raising attention to a previously overlooked community is especially important. It has long been taken by the mainstream research community that languages only exist in their standard forms while in fact, many people use their dialectal variations. In our case, **this is especially important as the dialect we are studying as pioneering researchers is used by a group that has long been historically treated unfairly. We are surprised to see this unfairness replicated in non-human entities that are so widely used in everyday activities, while this problem is paid very little attention to by mainstream AI research.** We think it is of significant importance to raise the question and provide an easy-to-use framework for people to develop better and fairer technologies for everyone.

That being said, **we think that our paper should not be valued less just because our finding that mainstream technology underperforms for minority groups is not surprising, or because we do not have any immediate explanation/solution.** Through scientific, rigorous, and objective verification, we have confirmed our disappointing finding. This finding also lays the foundation for future research aimed at mitigating the gap. After all, we all know language models are unfair, imperfect, and difficult to interpret. **We want to develop AI for the human good.** We hope reviewers and AC can consider our points raised here.

[1] Bender, E. M., Gebru, T., McMillan-Major, A., & Shmitchell, S. (2021, March). On the dangers of stochastic parrots: Can language models be too big?🦜. In Proceedings of the 2021 ACM conference on fairness, accountability, and transparency (pp. 610-623).

[2] Gupta, S., Shrivastava, V., Deshpande, A., Kalyan, A., Clark, P., Sabharwal, A., & Khot, T. Bias Runs Deep: Implicit Reasoning Biases in Persona-Assigned LLMs. In The Twelfth International Conference on Learning Representations.

---

### Note · Authors · 2024-12-02

I have read and agree with the venue's withdrawal policy on behalf of myself and my co-authors.